# Plastics—Villain or Hero? Polymers and Recycled Polymers in Mineral and Metallurgical Processing—A Review

**DOI:** 10.3390/ma12040655

**Published:** 2019-02-21

**Authors:** Sheila Devasahayam, R. K. Singh Raman, K. Chennakesavulu, Sankar Bhattacharya

**Affiliations:** 1Department of Chemical Engineering, Monash University, Clayton Campus, Victoria 3800, Australia; raman.singh@monash.edu (R.K.S.R.); sankar.bhattacharya@monash.edu (S.B.); 2Department of Mechanical & Aerospace Engineering, Monash University, Clayton Campus, Victoria 3800, Australia; 3Laboratory of Supramolecular Chemistry, Institut de Science et d’Ing’enierie Supramol’eculaires (ISIS), UMR 7006, CNRS, Universit’e de Strasbourg, 8 allee Gaspard Monge, 67000 Strasbourg, France; chennakesavulu@unistra.fr; 4Department of Chemistry & International Research Centre, Sathyabama Institute of Science and Technology (Deemed to be University), Chennai 600 119, India; chennanml@yahoo.com

**Keywords:** recycled polymers, virgin polymers, thermoplastics, thermosets, super absorbent polymers, silicones, iron and steel industries, clean coal, carbon sequestration, high temperature processing, ambient palletisation, emission and energy reductions

## Abstract

This review focusses on the use of recycled and virgin polymers in mineral and metallurgical processing, both high and ambient temperature processes, including novel applications. End of life applications of polymers as well as the utilisation of polymers during its life time in various applications are explored. The discussion includes applications in cleaner coal production, iron and steel production, iron ore palletisation, iron alloy manufacturing, manganese processing, E-wastes processing and carbon sequestration. The underlying principles of these applications are also explained. Advantages and disadvantages of using these polymers in terms of energy and emission reductions, reduction in non-renewables and dematerialisation are discussed. Influence of the polymers on controlling the evolution of micro and nanostructures in alloys and advanced materials is also considered.

## 1. Introduction

Polymers or macromolecules are derivatives of petroleum or natural gas characterized by long chain molecules, classified as elastomers (rubbers), plastics and fibers. They are the second-most important basic materials and are expected to exceed a global production of 420 million tonnes annually in 2020 [1]. The industrial, environmental and economic advantages of polymeric materials outscore the conventional materials throughout their life cycle, despite the end of life (EOL) considerations [1]. The fossil hydrocarbons used in the production of plastics are returned to the fuel cycle at the EOL (Figure 1) [2]. Table 1 shows the details of the global primary plastics production in 2015, according to the industrial sectors uses, lifetime distributions and the primary waste generation variance according to the industry sector usages [3]. Individual polymer types and their fate are listed in Table 2 according to their industrial applications [3]. 

### End of Life Options for Plastics

Concerns are mounting regarding the usage, disposal and accumulation of EOL plastics in landfills and in natural habitats. Ingestion or entanglement in plastic and leaching of chemicals from plastic products can affect wild and marine life and humans. Currently, there is a global consensus that plastics are a threat to the environment, prompting many governments to ban single use plastic products, including Australia, India, Kenya, U.K., Taiwan, Zimbabwe, Montreal (Canada), Malibu and Seattle (USA), Hamburg (Germany), France, Morocco, and Rwanda. 

About 4% of world oil production is used as a feedstock to make plastics and a similar amount is used as energy in the process. Waste plastics management practices for the EOL plastics globally are land filling, industrial energy recovery from municipal solid waste incineration (MSWI), pyrolysis and recycling. Used plastics can be recycled up to six times. If it did not make economic or environmental sense to recycle, then the energy can be recovered through energy from waste incineration [2]. Recycling reportedly has the lowest environmental impact on both global warming potential (GWP) and total energy use (TEU) in most cases [1]. Waste plastics recycling include mechanical (material) recycling, and pyrolysis or feedstock recycling:➢Land filling results in no net greenhouse gas (GHG) emissions, no GHG benefits and a waste of valuable resources [4].➢Mechanical recycling corresponds to the reprocessing of plastic waste to produce new products.➢Energy recovery option is used to generate heat, steam or electricity or as solid refuse fuel for co-fuelling of blast furnaces or cement kilns, or pyrolysis to diesel fuel or gasification. Highly mixed plastics, such as some electronic and electrical wastes and automotive shredder residues, are the candidates.➢Pyrolysis, a recycling technique, includes non-catalytic thermal or catalytic thermal methods. High-temperature pyrolysis of plastics is particularly advantageous for metallurgical processes running at very high temperatures, e.g., the Corex Process, blast furnace, HTK Process (HTK: Hochtemperatur-Konversionsverfahren, aka High-Temperature Conversion Process) of Voestalpine-Österreichische Mineralölverwaltung (VOEST-ÖMV), and the Thermoselect Process [5]. Unsorted plastic wastes are converted into fuels, monomers, and other valuable materials, such as liquid fuels, solids (char) and combustible gases, carbon, hydrogen chloride (from PVC) and bromine via thermal and catalytic cracking processes in the absence of oxygen at about 300° C to 500° C.
▪Gasification of plastics occurs approximately at 525–625 °C, much before the steel melting temperature [5].▪Chemical recycling routes can be broadly divided into thermochemical and catalytic conversion processes. Chemical or feedstock recycling breaks down heterogeneous, contaminated and hard-to-recycle waste polymers into their chemical constituents and converts them into monomers for new plastics or fuels, usually in blast furnaces.▪Thermal depolymerization breaks down waste plastics into crude oil products under high temperatures and pressure (>400 °C and 4 MPa) in the presence of water via hydrous pyrolysis. Long chain polymers depolymerise into short chain monomers. Small gaseous molecules like methane or carbon dioxide cannot be converted into oil using this process. In this process, organic toxins lose their toxicity due to the breaking down of chemical bonds. Heavy metals are eliminated from the samples as ionized or stable oxide forms. However, toxic by-products like furan and dioxin may be released in addition to methane and carbon dioxide. When applied to E-wastes, this process is effective in dealing with the plastics portion of the E-wastes but offers no method of bromine recovery. The solid fraction resulting from this process would have much higher concentrations of metals. Metal recovery from this metal-laden slag becomes complicated due to the high concentration of oxides, offering no overall increase in recycling efficiency. Thus, a thermal depolymerization process is not suitable for maximum metal extraction, or for increased recycling efficiency.

Recyclability and sorting of commodity thermoplastics is facilitated using a number identification code from 1 to 7 (Figure 2) [6]. It is economically viable to recycle up to polymer type 5 or polypropylene (PP). 

This review will focus on the specific applications, relevant plastic types, distinct properties of plastics or interactions among them and recycling and energy recovery of the relevant EOL polymers in mineral processing and metallurgy. 

## 2. Outline of Specific Applications of Polymers in Mineral and Metallurgical Processing

Applications of polymeric materials in mineral processing and metallurgy are determined by their pyrolysis products, bonding capacity with metal oxides as the alloying material and the super affinity to the water. In this context, this section discusses the relevant aspects of thermoplastics, thermosets, silicones and super absorbent gels. High-temperature pyrolysis of plastics is particularly advantageous in connection with metallurgical processes running at very high temperatures, e.g., the Corex process, the blast furnace, the recycling of the automotive wastes, the HTK Process (Hochtemperatur-Konversionsverfahren, aka High-Temperature Conversion Process) of VOEST-ÖMV, and the Thermoselect process [5]. This paper considers following applications of EOL and virgin plastics:Energy and fuel: Plastics have a high calorific value when compared with other materials (as they are essentially made from crude oil), making them a convenient energy source.Reductant: The CH_4_, CO, H_2_ and the char produced during the pyrolysis of the plastics can replace the coal-based coke as a reducing agent in blast furnace applications.Binder: Virgin thermoset polymers and inorganic silicones are adhesives and adhesion promotors (coupling agents). They form strong interfacial chemical bonds with transition metal oxide and impart high strength in the pellets using these thermosets as the binder. While epoxy thermoset resins lend high compression strength, the silicones, on the other hand, are easily compressible; nevertheless, they form stronger bonds with metal oxide and do not crumble. In addition, they also offer other advantages of the plastics as an energy source and as the reductants. They offer energy savings, better mechanical properties compared to the conventional binders, such as bentonite, which due to the high silica content, adds to the slag burden. Thermoset polymers offer high strength to the pellets under ambient conditions and can replace the conventional binders, such as bentonites, and organic binders in palletisation process [7].Alloying material: Virgin thermoset polymers and inorganic silicones also act as alloying agents in ferrosilicon and silicon carbide, and iron carbide. Silicones can substitute coke and quartz in iron alloy (ferrosilicon) production [8]. The viscous nature of the virgin thermosets provides for solid–liquid–gas reactions promoting good wetting between the reactants. This promotes the elemental silicon in silicones to react with the hematite to form ferrosilicon at much lower than the conventional temperatures. Similarly, the fillers in the epoxy resins easily form alloys with the hematite to produce SiC and FeC_8_ with nano- and microstructures.Carbon sequestration: Virgin thermosets can completely consume the carbon dioxide produced in a high carbon footprint reaction, such as magnesia and magnesium and clinker production, greatly reducing the global warming potential by up to 99%. This is achieved through the use of thermosets for carbon sequestration in the thermal decomposition of magnesite [9].Application in dewatering of coal: High moisture content in coal, in addition to detrimentally impacting the calorific value, also attracts high transportation tariffs and energy penalties when evaporating techniques are employed for dewatering. Superabsorbent polymers (SAPs) offer ambient dewatering of low-grade coals, greatly improving their calorific values and reducing the transport tariffs. Use of SAPs to dewater fine black coals, brown coals and tailings has been reported [10,11,12].

## 3. Polymers Used in Mineral Processing and Metallurgy

### 3.1. Recycled Thermoplastics and Thermosets

Pyrolysis of plastics, mainly the thermoplastics, and to an extent, thermosets are particularly advantageous in connection with metallurgical processes running at very high temperatures. Thermodynamic calculations of the pyrolysis of plastics show only two stable phases, solid carbon and the gas phase, in which two species, namely methane and hydrogen, dominate depending on the temperature; higher-grade hydrocarbons and pyrolysis oil are a possibly thermally unstable third phase [5]. The pyrolysis product and wetting characteristics of the polymers determine their usefulness, for example, in blast furnace applications, which vary for thermoplastics and the thermoset plastics. 

Pyrolysis of thermoplastics yield more volatiles and less carbon residue (less carbonization), whereas, thermosets yield more carbon residue. Thermoplastics also show better wetting characteristics whereas thermosets show little or no wetting characteristics as determined by their bonding characteristics.

Thermoplastics (Figure 3) undergo phase transformation and can be shaped and reshaped using heat. They undergo chemical [13,14], thermal, photo- or biodegradation and may be recast into a new product or repolymerized [15]. Amorphous thermoplastics undergo phase transformation from solid to the rubbery state above the glass transition temperatures. The ability of the thermoplastics to undergo phase transformation also determines their wetting characteristics. Crystalline or semicrystalline thermoplastics may not achieve the fluid or viscous state and they undergo thermal decomposition before they melt. Elastomers behave either like thermosets or as thermoplastics. 

Pyrolysis results in scission and/or cross-link products. Thermoplastics mainly undergo bond or chain scission reactions, involving breaking of the bonds (endothermic) of the long chain backbone, e.g., C–C and C–H bonds depending on the bond energies. Low thermal stability of polyethylene (PE) is attributed to the low bond energies of C–H and C–C bonds, i.e., 406 kJ/mol and 347 kJ/mol, respectively. Thermal processing of thermoplastics such as HDPE and LDPE at 550 °C shows no solid fraction (coke) (Table 3). The scission products of PE make up the liquid and gaseous fraction. Thermal stability significantly drops when more readily abstractable hydrogens from the carbon are present [16]. Presence of tertiary carbon atoms in the polymer backbone and the side group linked to the main chain also affect the thermal stability, e.g., polypropylene (PP) is less stable due to the presence of tertiary carbon atoms in the polymer backbone than polyethylenes (LDPE, HDPE or LLDPE). They are susceptible to pronounced thermo-oxidative degradation during melt processing and/or during use, as well as photo-oxidative degradation. 

Polyalphamethylstyrene undergoes “specific” chain scission with breakage at the chain ends to depolymerize into constituent monomers. Thermal stability of majority of polymers, e.g., PE, PP (polypropylene), PET (polyethylene terephthalate), PMMA and PC (polycarbonate) reduces in air due to the presence of oxygen as compared to an inert atmosphere. 

Thermoset plastics, e.g., epoxy-based products and composites, once manufactured, cannot be re-formed or re-used as they are intractable by design (Figure 4). Thermosets do not melt when heated unlike the thermoplastics. Therefore, they exhibit little or no wetting characteristics. However, thermoset polymers have calorific values that can be recovered as a source of energy for industrial processes. Most thermoset resins have a calorific value of approximately 30,000 kJ/kg [17]. Recycling options for thermoset composite materials include mechanical comminution to reduce the size of the scrap to produce filler materials, chemical degradation to break the polymeric matrix into simple chemical constituents and thermal degradation to break the scrap down into materials and energy [18]. While the thermoplastics degrade thermally via chain scission reactions, thermosets form cross-links, an important process for char forming which generates high molecular weight structures that are less volatile.

The cross-linking reaction is preceded by scission. Cross-linking in most cases is reflected in the strength, toughness and thermal stability of the polymers, which in turn depends on the molecular weight and the length of the polymer chains. High molecular weight and the size of the macromolecules and the intermolecular forces, e.g., hydrogen bonding enhances thermal stability. Cross-linking involves recombination of gases, liquids and solids resulting in non-condensable gas fractions, solid fractions including carbon, and liquid fractions (e.g., paraffins, olefins, naphthenes and aromatics such as benzene and toluene) [19] (Table 3, Figure 5). 

Cross-linked structures that possess high thermal stability require the simultaneous breakdown of several bonds in order to reduce the molecular weight facilitating char formation. Physical characteristics of the chars determine the rate of thermal decomposition of the remainder of the plastics. 

Char formation as depicted in Figure 5 occurs due to cross-linking or cyclization reactions producing cyclic structures, where two adjacent side groups react to form a bond to produce carbon-richer residue than the original polymer. For example, polyvinyl chloride undergoes cyclization reaction to form hydrogenated char (Reaction (1)) and poly (vinylidene chloride) forms pure carbonaceous char with an almost graphitic structure (Reaction (2)). The char further breaks down by scission at much high temperatures [20] resulting in weight loss in the char fraction.
–CH_2_–CHCl– → –CH=CH + HCl(1)
–CH_2_–CCl_2_– → –C=C– + 2HCl(2)

The yields from the pyrolysis of nonindustrial waste plastics are ≈20% coke, ≈40% tar and light oil, and ≈40% gases [21]. The liquid fraction consists of hydrocarbons in the gasoline range (C4–C12), diesel (C12–C23), kerosene (C10–C18) and motor oil (C23–C40) [22,23]. High temperatures decrease the yield of hydrogen, methane, acetylene and aromatic compounds, whereas low temperatures favour the generation of gaseous products [24]. The coke product or the solid fraction yield depends on the volatile content in the polymers, decreasing with increasing volatile content in the polymers. Similarly, the gas yield decreases as the carbon content increases (Table 4 and Table 5) [25]. Pyrolysis products of thermosets and their composites are given in Table 6 and Table 7 [26].

### 3.2. Virgin Thermoset Polymers

Epoxy thermoset polymers as adhesives find application in metallurgical processing [7]. Epoxies are adsorbed on the metal oxides, e.g., iron oxide forming interfacial bonds having energies ranging between 110 and 140 kJ·mol^−1^ [29]. 

Epoxy resins comprise at least two epoxy groups (a three-member ring) per molecule. Epoxy resins contain aliphatic, cycloaliphatic or aromatic backbones. The oxirane or the epoxy ring (Figure 6) can react with different functional groups, which makes the epoxy resins versatile. The reactions with curing agents result in three dimensional cross-links, which render the epoxies insoluble, and intractable thermosets. Epoxies have high adhesion to metals, glass and ceramic materials. They also possess high cohesive strength. Hence, the failure under stress does not occur at the interface or within the epoxy layer. They show high temperature resistance and low shrinkage as they do not release water or other reaction by products. 

### 3.3. Virgin Silicones (Organic–Inorganic Polymers)

Silicones (organic–inorganic polymers) (Figure 7) are a good source of reducing gases, such as CH_4_, H_2_ and carbon residues, and thus can substitute for metallurgical coke and silica in blast furnace operations. The current recycling practice of landfilling or catalytic depolymerisation of silicones [30,31] results in the loss of valuable resources, such as silica, methane, carbon and hydrogen as alloying or reducing agents. Contamination of methane biogas by volatile organosilicon materials, disposed at the landfill sites, is a growing environmental issue. Noxious odours and extreme temperatures limit the recyclability of silicones. 

There is an opportunity to use silicones as coupling agents or adhesion promotors in metallurgical processing. Silicones form strong intermolecular interactions and chemical bonds with inorganic materials, such as glass, silica, alumina, talc, clay, mica, aluminium, iron, titanium oxide, zinc oxide, iron oxide, graphite, carbon black and calcium carbonate, showing improved wettability, compatibility, heat resistance up to 315 °C, weatherability and moisture resistance [32]. Silicones, composed of a Si–O linkage, have a high bond energy of 444 kJ·mol^−1^. The Si–C bond has a bond energy of 318 kJ·mol^−1^, slightly lower than a C–C bond, while the Si–Si bond is weak (193 kJ·mol^−1^). An Si–H bond (bond energy of 320 kJ·mol^−1^) is more reactive than the C–H bond of hydrocarbons (bond energy of 416 kJ·mol^−1^). The Si–O bonds in silicones are strongly polarised, compared to C–O bonds in organic polymers, with stronger intermolecular interactions [33]. The epoxy-metal oxide bond strength is ≈140 kJ·mol^−1^ compared to that of siloxane-metal oxide (≈450 kJ·mol^−1^). The Si–O bond in silicones can replace the metal–oxygen–hydrogen bond or even the carbon–oxygen–metal bond easily through strong surface bonds [34] resulting in a dense cross-linked network that is interwoven with terminal chemical functionality. The bond energy of such an interaction in a glass-silanol system (Si–O–Si) is ≈450 kJ·mol^−1^. Bond strengths at the interface are listed in Table 8 [35]. Silicones show increased wetting with the transition metal oxides, e.g., iron ore (iron oxide) compared to the carbon-based polymers, and increased mechanical strength. Iron oxides render the silicones with increased thermal stability at elevated temperatures [36]. 

Use of virgin silicones provide a reduced carbon footprint for many essential products and services during their life cycle as per the studies conducted in North America, Europe and Japan. Use of virgin silicones, siloxanes and silanes in energy enabling technologies, energy and materials results in energy savings and greenhouse gas (GHG) emissions reductions. The benefit/impact ratio = 9, calculated by dividing the benefits by the impacts from production and end-of-life; when using silicones, siloxanes and silanes outweighs the impacts of production and end-of-life disposal (Figure 8). The benefit/impact ratio greater than 1 indicates that the use of the silicone product is advantageous in terms of GHG emissions [37]. The CO_2_ emission cuts realized in the three regions amount to an estimated 54 million tons per year [37]. 

### 3.4. Superabsorbent Polymers 

Super absorbent polymers (SAPs) (Figure 9) offer a low energy alternative to de-water brown and fine black coal particles and the tailings [10,11,38,39,40]. SAPs can be cationic, anionic or non-ionic and comprise high molecular weight cross-linked hydrophilic polymers that absorb several tens to hundreds of times their individual mass of water as they expand in size, but still preserve distinct particle identity.

The amount of water that a specific SAP can absorb depends on its chemical composition and morphology as well as the quality of absorbed water [41], particularly with respect to the presence of ionic salts. The recycled water used in mining plant operations generally contains a significant concentration of salts [42] that needs to be managed to maximize the polymeric absorption potential. Highly anionic polymers are most effective in soft waters, having low mineral content comprising only sodium ions and free from dissolved calcium and magnesium salts. In hard waters, having high mineral content of dissolved magnesium and calcium salts, polymers with a lower anionic content perform better. pH-sensitive SAPs (cross-linked, partially hydrolyzed polyacrylamide gels) absorb moisture and swell at neutral pH. The swollen gel is separated from the dewatered coal by filtering. Subsequently the pH of the swollen gel is lowered to release water towards regeneration of the SAPs for reuse. Temperature-sensitive SAPs (cross-linked poly(*N*-isopropylacrylamide) and cross-linked copolymers of 97% *N*,*N*-diethylacrylamide and 3% sodium methacrylate) absorb water and swell at 25 °C and are regenerated by heating to 33 °C and 55 °C, respectively, when they collapse releasing the water.

## 4. Mineral and Metallurgical Processes Using Recycled and Virgin Polymers

This section discusses in detail the specific applications of the polymers in mineral and metallurgical processing outlined in Section 2. 

### 4.1. Dewatering of Coal

Dewatering of fine coal, an important aspect of coal cleaning, adds significant cost to the price of clean coal. Victoria has a reserve of 430 billion tons of brown coals, contributing to 92% of electricity generation in Victoria. However, it has a high moisture content that reduces its net calorific value, and thus is a major contributor to total GHG emissions, besides adding to high transportation costs and energy penalties. The high heat capacity of water inhibits use of evaporative drying technologies due to high energy penalties. Capital expenditure (CapEx) of conventional dewatering techniques are: for the centrifuge technique: < Au$50,000/t/hr/rel. reduction in total moisture content (TMC), for vacuum filtering: Au$450,000/t/hr/rel.reduction in TMC, for pressure filter technique: <Au$350,000/t/hr/rel. reduction in TMC, and for briquetting: <Au$350,000/t/hr/rel. reduction in TMC [43]. 

Types of moisture distribution within fine coals are (Figure 10):Inter-particle capillary and/or pendular moisture, which can be removed by mechanical techniques broadly classified as centrifugal, vacuum or pressure dewatering processes.Surface and intra-particle moisture (held by capillary forces or by chemical adsorption within the macro and micro pores of the particles), which can be removed by thermal drying.Chemical water, which is held in chemical combination with certain mineral matter in the coal and may only be released during combustion or gasification process.

Dzinomwa, et al. (1997) [10] employed SAPs to de-water fine black coal particles, which revealed advantages over the alternative non-evaporative drying technologies. SAPs offered an alternative lower energy osmotic water removal approach to the conventional thermal technologies. Table 9 compares the % water absorption capacity of different materials including the SAPs. 

Victorian brown coals have high oxygen and moisture contents and hence low calorific value (≈8000 kJ/kg) before drying or dewatering. However, on drying or dewatering, the calorific value is high (≈28,000 kJ/kg). Osmotic dewatering of brown coals using anionic SAP carried out under ambient conditions has no associated energy penalties. Approximately 57% of water is removed within four hours of SAP contact under ambient conditions [11,45]. This process, in addition to enhancing the calorific value of low-grade coals, greatly reduces the penalties associated with transportation. This process offers a low carbon, clean energy footprint technology to address energy problems. Though SAPs are about 100 times more expensive (1000 Au$/tonne) than the coal, the pH and the temperature sensitive SAPS are regenerated, which makes it cost effective. The pH-sensitive SAPs are regenerated at room temperatures, while the temperature-sensitive SAPs are regenerated at 37 °C and 55 °C, making them energy efficient compared to the evaporative dewatering. Moreover, the spontaneous combustion associated with heavily oxidized low rank coals, and the loss of volatiles associated with high temperature drying, can be mitigated using the SAPs. This technology can be extended to dewatering tailings/waste sludges. 

Mechanism of water removal is as follows: dry SAP having higher concentrations of “free” COO– groups than the coal removes moisture preferentially from coal through osmosis [46]. Factors affecting the capacity of SAP to absorb water are as follows:➢Swelling properties (attributed to presence of hydrophilic groups in the network).➢Cross-linking density (generally higher molecular weight with lower cross-linking densities exhibits higher absorption capacities).➢Structural integrity (high cross-linking density is crucial to retain the structural integrity of the polymer loaded with moisture as the high cross-linking density offers high mechanical strength).➢The “availability” of target water, i.e., how tightly it is bound to the drying substrate.

Figure 11 shows the mechanism of water absorption in SAP. The initial diffusion of water inside hydrophilic SAP causes ionization of neutralized acrylate groups into negative carboxylate ions and positive sodium ions. Negative electrical charge along the SAP backbone causes mutual repulsion of carboxylate ions, increasing the osmotic pressure inside the gel, thereby resulting in expansion and swelling of the SAP chains due to absorbed water. Finally, cross-links between chains inhibit solubilisation of SAP (in water), thus, governing the extent of swelling or absorption by restricting infinite swelling. Incorporation of nanofillers like graphene oxides or clay having laminar structures or pozzolanic flyash having high affinity to moisture within the SAP network (SAP-nanocomposites) can enhance the water-removing capacity of the SAPs while providing the necessary structural reinforcement against infinite swelling [47].

### 4.2. Coke Oven

Coke is utilised as an iron ore-reducing agent in blast furnaces, hydrocarbon oil as raw materials for the chemical industry and the gas as a fuel for power plants. Caking coals, e.g., many bituminous coals, when heated, exhibit thermoplastic effects, i.e., they soften and form a plastic mass that swells and resolidifies into a porous solid. Strongly caking coals, which yield a solid product (coke) with properties suitable for use in a blast furnace, are called coking coals. 

Japan uses waste plastics to produce coke, coke oven gas (COG) and hydrocarbon oil in a coke oven [48,49]. Nippon Steel & Sumitomo Metal Corporation (NSSMC) successfully developed a commercial-scale waste plastic-recycling process in coke ovens that treats ≈200,000 tons per year of waste plastic [50]. In coke ovens, waste plastics are not burnt but decomposed to produce chemically useful materials that are easily recovered (Table 10). HCl gas from the carbonization of polyvinylchloride wastes is trapped by ammonia liquor and used for cooling hot COG [50].

The effect of plastic addition on the coal-caking property varies with the type of plastics [51]. The addition of PE and PVC had only a small effect on the coal-caking property and coke strength (Figure 12 and Figure 13). While PE addition increases coke strength, addition of PS and PET inhibits coal expansion and fusion and deteriorates the coke strength. When the free radicals formed during PS or PET thermal decomposition removes hydrogen from the coal, the coal-caking property decreased. Coke strength is determined by the total surface area of the plastic particles added. Increasing the plastic particle size and density of the waste plastic particles decreases the specific surface area and increases coke strength. The optimum size of the plastic agglomerates used by NSSMC is in the range of 20–30 mm. A denser plastic agglomerate allows for more waste plastics to be added to coal without affecting the coke strength.

Coke is produced in a coke oven by carbonizing, i.e., heat treating the coal without oxygen at a high temperature. Coke can be produced via the chemical feedstock recycling of waste plastics in the coke oven [50]. When agglomerated waste plastics blended with coal at the blending ratio of about 1 wt. % are charged into the coke oven they are converted into coke, COG and hydrocarbon oil like coal in the coking chamber. In this process, carbon and hydrogen contained in the waste plastics are converted to 20 wt. % coke, 40 wt. % hydrocarbon oil (tar and light oil) and 40 wt. % coke oven gas (COG). The yield depends upon the type of plastic used (Figure 12). The reduction potentials of CO_2_ emissions by coke oven chemical feedstock recycling depends on the calorific values and the coke product yields of each plastic resin. For example, in the case of coke oven chemical feedstock recycling, the reduction potential of PS and PP is larger than that of PE [25].

Energy recovery from waste plastics and feedstock recycling are the underlying principles of the coke oven process.

### 4.3. Iron and Steel Making

Use of both recycled and the virgin polymers in iron and steel making is discussed in this section.

#### 4.3.1. Recycled Polymers

Annual production of steel is about 1.3 billion tonnes worldwide [52]. The iron- and steelmaking industry consumes 5% of the world’s total energy, being one of the most energy-intensive industries, with an annual energy consumption of about 25.8 EJ in 2007 [53]. The iron and steel industry is one of the biggest industrial CO_2_ emitters. Globally, between 4 and 7% of the anthropogenic CO_2_ emissions originate from this industry [54]. 

Iron ore is reduced with coke in a blast furnace, and liquid steel in a basic oxygen furnace (BOF) is produced from the reduced hot metal (pig iron) mixed with 30% steel scrap. In the electric arc furnace (EAF) process, the iron input is typically in the form of scrap, direct reduced iron (DRI) and cast iron. To produce 1 ton of hot metal, approximately 370 kg of coke plus 90 kg of heavy oil are used depending on the quality of the iron ore and the process used. The 90 kg of heavy oil can be partly substituted with up to 70 kg of plastics waste [4,54].

The high temperature processability of the thermoplastics (Figure 3) finds applications in blast furnaces, especially in iron and steel processing [55,56]. The Japanese steel industry developed commercial waste plastic feedstock recycling processes using its blast furnace iron- and steelmaking (electric arc furnaces) processes. In the blast furnaces, finely pulverized dechlorinated waste plastics and pulverized coal are injected into the blast furnace through tuyeres as a substitute for coke to produce pig iron [33,57,58]. The blast furnace feedstock recycling substitution rate is 1.1 (1.1 ton of coke is substituted by 1 ton of waste plastics) [25]. The substitution rate of plastic resins is directly correlated to their carbon content or calorific value. Plastics have higher calorific value compared to the coke (28,000–31,000 kJ/kg) (Table 5). The calorific value of gases (blast furnace gases) increases with all the plastic resins because of the higher calorific values of plastics than that of coke. Pyrolysis of thermoplastics yields low carbon residue and high volatiles at ironmaking temperatures.

Plastics contain nearly 3 times more hydrogen than the pulverized coal and use of plastics increases the H_2_ content within and in the off gas exiting the blast furnace. The increased bosh gas H_2_ reduces the pressure drop to allow a greater gas flow for the same pressure by decreasing the bosh gas density. Injecting waste plastics lowers CO_2_ emissions by about 30% in comparison to coke and coal [59]. 

The CO_2_ emission reduction potential in blast furnace feedstock recycling depends on the calorific value and the carbon and hydrogen content of the individual plastic resins. Thus, PE has the largest CO_2_ emissions reduction potential. The CO_2_ emissions reduction potential of PP is less than those of PE and PS, while PET has negative CO_2_ emissions reduction potential. This leads to a difference in the coke substitution effect by each plastic resin [25]. The measured carbon pick-up after 2 min of reaction in reduced iron ore using metallurgical coke is approximately 0.08 wt. %, whereas 100% thermoplastics, such as PET and HDPE, showed an increase in carbon pick-up by >2 wt. % [60]. The rate of carburization is controlled by the dissociation of CH_4_ from waste plastics on the liquid iron surface (Reaction (3)). The presence of H_2_ was found to aid the carburization, probably through the removal of adsorbed oxygen from the metal surfaces [61]. Carbon dissolution is reported to be: for graphite ≈3%, pet coke ≈4.2 %, charcoal ≈2%, coal and nut coke ≈1.5% [62].
CH_4_ = C (in Fe) + 2H_2_(3)

In EAF steel making, replacing ≈30 % of the coke and coal by polyethylene wastes resulted in energy savings up to ≈12 kWh/t of plastic charge [63]. Tyre injection trials at OneSteel’s Sydney steel mill, Australia, saw reduction in electrical energy consumption from 1.526 MJ/kg to 1.483 MJ/kg billet, (i.e., 0.043 MJ/kg savings in energy), the amount of carbon injectant reduced from 464 kg/heat to 406 kg/heat, and the improved number of liquid tonnes per power-on time minute from 2.12 to 2.2 t/min [64].

While extensive use of thermoplastics as a partial substitute for coke in iron and steel industries has been reported, the use of thermoset plastics has not been reported to the same extent. Dhunna et al. (2014) [65] studied the use of raw Bakelite, a thermoset, as well as the char of the Bakelite (prepared by heating it to 1450 °C for 20 min) for iron ore reduction. It is found that when raw Bakelite was used, a clear separation of metal and slag phases could not be achieved due to low carbon uptake (0.23%). On the other hand, the use of Bakelite char (char obtained by heat treatment of Bakelite at 1450 °C) for iron ore reduction resulted in a clear separation of metal and slag phases due to higher carbon uptake (2.01%). Also, reduction efficiency was slightly higher (≈6%) for the Bakelite char compared to the raw Bakelite [65]. Raw Bakelite-based pellets showed an 83.9% reduction of iron oxide, whereas Bakelite-char pellets showed 90.12% metallization after 20 min of reaction. A higher carbon dissolution is observed for the Bakelite chars due to the change in microporosity generated as a result of evolving volatiles and resulting in increased surface area and decreased pore size as well as the graphitic carbon and low ash content. The surface area reduced quickly, from ≈325 to 25 m^2^/g in two minutes of the reaction [28]. 

The above study suggests thermoplastics perform better in terms of reduction and carbon dissolution compared to the thermosets. The ability of the thermoplastics to melt or undergo phase transformation from solid to the rubbery state above the glass transition temperatures offers a different mechanism compared to the thermosets, which cannot undergo phase transformation. A solid–liquid–gas system provided by thermoplastics promote intimate contact and wetting between the reactants, which the thermosets are incapable of. However, the volatiles (CH_4_ and H_2_) from the Bakelite thermosets contribute to the reduction efficiency. High temperature char formation from Bakelite helps increase the carbon dissolution in the liquid iron phase [65]. However, using Bakelite char eliminates the benefits of the volatile reductant gases, namely the CH_4_ and H_2_, in addition to being a high temperature (energy intensive, ≈1450 °C) process. It is not clear why the CH_4_ and H_2_ evolved from the Bakelite does not help carbon dissolution nor why only Bakelite char contributes to the carbon dissolution and not raw Bakelite (Reaction (4)).
CH_4_ (g) = C (s) +H_2_ (g)(4)

Raw Bakelite mixed with the hematite under the reduction reaction conditions should form chars and contribute to the carbon dissolution. A carbothermal reduction process requires intimate contact between a solid oxide and a carbonaceous material [66]. However, the study has used different particle sizes of the Bakelite: raw Bakelite is 125 µm, whereas Bakelite char is further homogenized in a roller crusher to a smaller size. The lack of intimate contact between the raw Bakelite and the hematite as opposed to better contact between the Bakelite char and hematite seem to contribute to the difference in the reaction efficiencies in the two systems. 

In blast furnaces, energy recovery from waste plastics and feedstock recycling of the mixed plastic wastes are the underlying principles to reduce iron ore into iron, replacing fossil resources like heavy fuel oil, coke, coal or gas, and other valuable resources.

#### 4.3.2. Virgin Thermoset Polymer

##### Iron Ore Palletisation 

Palletisation process improves physical properties of low-grade fine ores, imparting high and uniform mechanical strength to withstand thermal stress in reducing atmosphere in blast furnace, and show resistance to disintegration. This makes transportation over long distances feasible without generation of fines. Waste plastics can be used in palletisation, but strength and hardness of plastics can be an issue in blast furnaces. Low strength agglomerated plastics are easily disintegrated during transport, which may lead to blockage problems and combustion, lowering combustion efficiency [55]. 

Bentonite, the most used conventional binder for palletisation with >50% SiO_2_ is detrimental for iron ore concentrate and can decrease the iron content by 0.6–0.7 wt. % in the pellets [67,68]. The energy consumption is increased by 30 kWh/t in direct reduced iron (DRIs) for every percent of acid gangue addition [69]. Silica-free organic binders, such as carboxymethyl cellulose (CMC), starch and dextrin are alternative binders to bentonite. However, organic binders fail to impart necessary strength to the pre-heated and fired pellets due to reduced slag-oxide bonding despite producing good quality wet and dry pellets. This is because they are burnt out at relatively low temperatures (<250 °C) [70] with little residue to bond with iron oxide grains at higher induration temperatures. For these reasons, improving the strengths of pre-heated and fired pellets produced with organic binders has assumed great significance in recent years.

Use of recycled thermoset plastics in the ironmaking process may not help carburation unless converted to char at 1450 °C (i.e., a high temperature and an energy intensive process) due to their inability to reform to provide the required intimate contact or wetting. Use of virgin thermoset polymers, on the other hand, serves as a binder, reductant, alloying material and fuel, thereby, minimising the need for non-renewable resources such as coke as the reductant and fuel. Iron ore palletisation using epoxy as the binder instead of conventional bentonite or organic binders shows many desirable features [7]. 

Thermoset polymers like epoxies are low molecular weight viscous polymers. During chemical reactions with curing agents (i.e., hardeners), they undergo further polymerisation and phase transformation from a viscous state to a dry, solid three-dimensional cross-linked molecular network. The iron ore pellets produced under ambient conditions using the thermoset epoxy resin showed a high compression stress of 4 kN without any heat treatment compared to the industry requirement (2.5 kN) of an indurated pellet processed at 1200–1350 °C. The pellets are reported to be non-abrasive, non-sticky and devoid of undesirable fines. The process spells cost, energy and emission reductions for iron and steel industries [7]. The study shows that the use of commercial epoxy resin, epiglue and silicones eliminates the need to use coke or high pure quartz Previous studies using recycled thermosets mainly involved a solid–gas reaction for coke/char-metal oxide. Virgin thermosets offer a different wetting mechanism, involving a solid–liquid–gas reaction that provides a better intimate contact, reaction efficiency and high carburation. A high carbon uptake of up to 3% is observed even in the absence of coke or other carbonaceous materials. 

Energy consumption during heat induration of magnetite concentrates ranges from 320,000 kJ/t to 740,000 kJ/t pellets, and it is higher for higher non-magnetite ores due to the absence of exothermic and oxidation reactions. Cold-bonded pellets using epoxy binders, being a low temperature alternative to heat-indurated pellets, consume less energy for bonding self-reducing pellets and for refractory ores that are difficult to heat treat. The estimated cold-bonding capital and operating costs is 2/3 that of heat induration. 

Use of virgin epoxy resin promotes dematerialization in terms of carbonaceous materials and the binder. The conventional iron ore reduction/palletization requires at least 1:3 Fe_2_O_3_ to C and a binder. However, in a recent study, the binder acts as the reductant and fuel at ≈1:0.3 Fe_2_O_3_ to C, i.e., at least 10 times less carbonaceous material [7]. The pyrolysis of resins also provides for reducing atmosphere of H_2_ and CO from the methane generated at the reaction temperatures, which aid the reduction reaction, as well as the solid carbon residue [5]. The high compression strength and the porosity of the pellets as a potential spacer in blast furnace application that facilitate gas transportation could diminish the use of non-renewable coke by more than the current 40%.

The mechanism involved in producing pellets of high strength can be explained as follows: Viscous adhesive (epoxy resin) is capable of dissolving and diffusing into iron oxide [71,72]. The diffusion depends on the affinity of the epoxy to the iron ore and vice versa, resulting in a multilayer interphase (0.5 nm to 10 µm thick) [73] controlled by the cure conditions. At longer liquid–solid contact times, and with a more gradual rise in the cure temperatures, tri-layered rather than bi-layered systems are formed. A tri-layer system shows a gradient and a higher Young’s modulus than the bilayer system because of the formation of fibre-like crystallites within the organic layer [74]. Mechanical properties of the bi-layer system are like those of the bulk. The addition of fillers to the organic resin increases the modulus and the load needed to propagate a crack [75,76] as they can act like little springs, making crack propagation difficult. The presence of nanocrystals formed during the curing of the adhesive layer, due to dissolution and recrystallization of the pigments and fillers, result in increased interfacial toughness [77]. The nanoparticle–polymer interaction is stronger because of increased surface area. Addition of ethylene oxide can increase the toughness by increasing the wettability between the fillers and the polymer interphase [78,79,80]. The excess epoxy monomer in the primer and the adhesive can thus increase the toughness of the interphase layer by softening the filler–polymer interface. An increase in the molecular weight due to the cross-linking of the polymer as the cure proceeds can suppress cavitation by increasing the cohesive strength of the amorphous phase. When the crystallites in a polymer, semicrystalline polymer, or crystalline polymer orient themselves in the loading direction, they exhibit not only a high elastic modulus but also a toughness approximately 15 times higher [81]. The process can be made greener using bio-renewable epoxy resin [82]. 

##### Alloy Formation

Silicon carbide (SiC) and ferrosilicon (FeSi) find application in the clean energy systems such as photovoltaic industries, and economical production of high purity hydrogen. It is estimated that the global demand for silicones will rise 6% per year to 2.4 MMT (million metric tonnes) in 2018. SiC is produced from petroleum coke and a silica or quartz sand at high temperatures (1700–2500 °C). FeSi is produced industrially via the carbo-thermic reduction of silicon dioxide (SiO_2_) with carbon in the presence of iron ore, scrap iron, mill scale or other source of iron at temperatures greater than 1800 °C.

Silicones are used in the production of ferrosilicon and the epoxy resin (Epiglue) having silica as the filler are used for producing the silicon carbide [8]. The study demonstrates the production of ferrosilicon/silicon carbide using only the hematite and the polymer resins (virgin polymers) as fuel/reductant and alloying material. The system has the added benefits of greatly reduced emissions, process temperature, non-renewable resources and cost of production is reported [7,8]. The process does not require the use of high-purity quartz or metallurgical coke. Formation of SiC and the ferrosilicon in presence of these viscous virgin polymer resins occurs at 1550 °C as opposed to the conventional temperatures of ≈2000 °C that spells energy savings and emission reductions. 

A carbothermal reduction process requires intimate contact between a solid oxide and a carbonaceous material [67] facilitated by the interfacial chemical bonding and the cross-linked network between the reactants, which inhibits the escape of SiO, an intermediate product in the SiC formation, from the reaction site. 

Presence of methane is reported to increase the SiC content, when supplied externally together with hydrogen. However, when C/SiO_2_ is less than the stoichiometric requirement of 3, Si is lost as SiO. While it is expected the carbon deficit could be compensated by introducing methane gas to increase the SiC content, it was observed [67] that the intermediate product, SiO vapor, escaped from the reaction zone as fume during the direct reduction of quartz or amorphous silica powder by methane without conversion to SiC. Carbon formed on the reactant surface during methane cracking (Reaction (4)) hinders the progress of reduction reaction by preventing carbon transfer to the interior reaction sites for further reduction when the methane content supplied externally is >2 vol %. Formation of carbon can be controlled by the increased external supply of hydrogen in the gas mixture. 

However, a solid-liquid-gas reaction facilitated by the viscous virgin polymer resin with the reactants can ensure intimate contact between the SiO and the insitu production of methane can increase the SiC content [8]. The pyrolysis of the polymer resin, being the source of CH_4_, H_2_, C and Si, in addition to facilitating the reduction of the iron oxide, aid the dissolution of carbon and silicon in the iron matrix to form SiC, FeSi and FeC_8_. Important technical aspects in this context are [8]:SiC and FeSi formation does not require any high-purity quartz or coke.Solid-liquid-gas reactions facilitated by the viscous polymer resin help achieve high reaction efficiency.Resources such as H_2_, CH_4_, C and Si produced in situ are derived from virgin polymers resins.There is a potential to recover these resources from feed stock recycling of thermosets and inorganic–organic polymers like silicones. However, it is necessary to make sure intimate contacts between the reactants are facilitated.Iron oxide–polymer composites as potential replacement for coke as spacer.Nano- and microstructured alloy phases are formed.

The reaction mechanism of alloy formation can be explained as follows: the virgin epoxies and the silicones in the fluid or viscous form prior to curing facilitate solid–liquid–gas system providing intimate contact between the reactants. They thoroughly wet the reactants and form interfacial covalent bond between the reactants upon curing (Figure 14a,b) [29,83]. Siloxanes, as a coupling agent, provides a stronger covalent interaction between the iron oxide and the Si–O bond and the pyrolytic products retained within matrices and form reduction products rather than leave the system without reacting, e.g., SiO. Where waste silicones and the thermoset resins are used, they open the pathway for feedstock recycling; however, use of bio-renewable epoxy resins [82], or recyclable epoxies, using Connora Technologies’ Recyclamine hardeners that can be re-used or repurposed are recommended [8,84]. GHG benefits of using virgin silicones (Figure 8) should also be considered.

### 4.4. Manganese Processing

The reduction of MnO in slag by blends of coke and high-density polyethylene (HDPE) shows improved wettability and reduction efficiency compared to reduction by pure coke. Increasing HDPE content resulted in further improvement in reduction and wettability and allowed for the near complete reduction of MnO and partial reduction of SiO_2_ to Si from the slag [85]. Decrease in CO_2_ emission as well as the evolution of CO and CH_4_ indicated the thermal decomposition of the polymer to liberate significant amounts of CH_4_ [85]. The production of manganese from MnO_2_ using mechanically recycled PP utilisation in the pre-reduction step (MnO_2_ → Mn_2_O_3_/Mn_3_O_4_ → MnO) is demonstrated in Reference [86]. However, the last transformation step MnO → Mn requires both solid carbon and CH_4_ as reductants [87]. Reduction of MnO from SiMn slag using metallurgical coke and its blends with HDPE at 10% has been reported [87].

Mechanical recycling of PP and the feedstock recycling of HDPE and the PP are employed in the manganese processing.

### 4.5. E-Wastes Processing

It is predicted that 49.8 million metric tons of e-waste will be produced globally in 2018 and 52.2 million metric tons by 2021 [88]. Potential value of raw materials in e-waste in 2016 is 55 billion Euros [88]. The main concern relating to e-waste disposal is the leaching of hazardous metals, such as chromium, mercury and cadmium, into the landfill as well as the surrounding aquifers, impacting on valuable water resources [89,90]. 

E-waste is a source for toxic compounds such as antimony, lead, mercury, cadmium and brominated flame retardants; reclaimable materials include a mixture of metals such as copper, aluminium, steel, gold, silver and platinum group metals and various types of plastics and ceramics. The material composition of the electronics is given in Table 11. The major economic driver for recycling e-waste is the recovery of precious metals, resource scarcity and depletion and high costs of critical elements, such as europium and neodymium, essential to high-tech manufacturing [91,92]. E-wastes recycling is viewed as an alternate materials resource and an important income generating activity as well as new economy [91]. 

E-wastes are processed in large-scale integrated energy- and resource-efficient smelters/plants operated by multinational companies, such as Boliden Minerals (Sweden), Norddeutsche Affinerie (Germany), Union Minière (Belgium), Noranda (Canada) and Outokumpu (Finland). They use a combination of different recovery methods such as mechanical recycling, feedstock recycling and energy recovery. Integrated processing of e-wastes increases eco-efficiency, which otherwise could be lost in the isolated steps. Main features of integrated processing of e-wastes are:The integrated furnace has necessary installations to collect the off-gases.The plastic fraction serves as a replacement of coke either partially or fully as a reducing agent as well as the fuel.The rapid heating above 1000 °C will contain the emissions of Dioxin/Furans within the acceptable limits.The copper/nickel/lead base metal metallurgy allows the precious and the lead special metals to be collected as slag.The smelter separates precious metals in a copper bullion. Other metals concentrate in a lead slag are further processed at base metals operations.The ceramics and the glass are recovered as a slag.

At Noranda reactor recycling facility, about 100,000 tons of e-waste is processed for metal recovery. Energy usage in the Noranda reactor is reduced due to the combustion of plastics in the e-waste feedstock [94]. UMICORE (Hoboken, Belgium) has an integrated metals smelter and refinery where, precious and other non-ferrous metals are recovered from e-waste [95]. Umicore Precious Metals Refining, one of the world’s largest recycler of precious metals from e-waste (Figure 15), treats roughly 250,000 metric tons annually, 10% of which is electronic waste. The major steps in Umicore’s process are collection, dismantling, shredding/pre-processing, and end processing. Umicore’s integrated smelting/refining process recovers Sn, Pb, In, etc., from mobile phones. In addition, their plastic components cannot be feasibly recycled due to the mix of flame retardants, pigments and mixed types of plastics. Umicore can process mixed plastic fraction in order to recover copper, precious metals, etc., while at the same time using the organic content of the plastics to replace the fuel and reductant used in this process [95]. This is only possible because of the sophisticated off-gas and water purification equipment installed at the Hoboken facility, safely preventing the emission of dioxins, furans or other harmful elements. Up to a certain range, this process can handle bromine and other halogens, which are often present in these plastic fractions as flame retardants.

The Kaldo furnace can recover metals from more than 100,000 tons of secondary raw materials from cable scrap and printed circuit boards. Plastics are used as fuel and easily oxidisable impurities are dissolved in a liquid slag and precious and base metals are collected as an alloy or “Matte” (liquid sulphides) [96]. The role of plastics from cable insulation is to supply process heat for the smelting operation. PVC and cross-linked low-density polyethylene in the cable scrap, as well as the thermosetting resin in the printed circuit boards, perform this very valuable function in the Cu recovery process. Their high heat value provides most of the process heat needed for secondary raw material smelting to produce prime grade copper.

Boliden’s Rönnskär smelter, in northern Sweden (one of the largest recyclers of electronic material) has been recycling various waste materials since the 1960s. The smelter’s annual capacity for recycling electrical material is 120,000 tonnes, sourced primarily from Europe. Plastic in the e-waste acts as a source of energy during smelting and generates steam that is converted into electricity or district heating. Boliden has the technical capability to take the entire shredded PC scrap into their zinc fuming plant without extensive dismantling. 

#### Dioxins/Furan Management

The zinc fuming furnace can effectively reduce the dioxins and furan content when processing the E-wastes [96]. The high risk associated with Polyhalogenated Dibenzo-p-Dioxins and Dibenzofurans, PXDD/Fs presence in mechanical recycling of WEEE, in relation to handling and exposure is minimised in feedstock recycling. The process essentially represents an effective sink for dioxins and heavy metals. The average PXDD/Fs content of the E-wastes processed in the zinc fuming furnace must meet the “German Regulations for Hazardous Materials” [97] and “The German Chemical Banning Ordinance” [98]. 

The zinc fuming plant consists of three separate processes, namely, the fuming furnace, settling furnace and the clinker furnace. The reduction process in the fuming furnace reduces metal oxides such as magnetite, lead oxide, zinc oxide, lead and arsenic, and extracts halogens from the slag. The resulting mixture of oxides is de-halogenated in the clinker furnace. The subsequent process of halogen removal in the clinker furnace can handle increased load of halogens from the steel making dust and other feed streams. Halogen removal is carried out in the clinker furnace by re-heating the raw fume together with coke additions to approximately 1200 °C. The zinc thus produced show dioxins values below the detection limits. The total halogen content liberated in the fuming plant is neutralised in the gas cleaning section [96]. The total halogenated dioxin/furan mass balance shows a destruction over 98% of all micro-organic compounds from this family. From the total input of 6.4 g per batch, only 0.104 g per batch leaves the fuming plant, evenly split between the gaseous emissions and the amount staying with the raw fume (Table 12). The raw fume when treated a second time at high temperature in the downstream kiln, destroys residual PXDD/Fs on the solid, when >99% destruction efficiency is achieved [96]. 

The heat balance analysis of plastic substitution from e-waste in comparison with normal operations using coke suggests plastics are used as the chemical feed stock, where carbon and hydrogen from the plastics substituted for the carbon and hydrogen provided by the coal for reducing the slag. This conclusion is arrived at by comparing the fuming speed, the generation of steam, and the specific energy as shown in Figure 16. A progressive reduction in the average coal consumption when plastics take part in the reduction of zinc oxide from the slag is seen in Figure 17 [96].

Mechanisms of e-waste recycling involves combination of recovery methods such as mechanical recycling, feedstock recycling, thermal depolymerization and energy recovery. Plastic components of e-waste cannot be easily recycled due to the mix of flame retardants, pigments and mixed types of plastics. The most common outlets for these mixed fractions are landfill and incineration plants, which are becoming increasingly restricted and costly given recent legislative developments. 

Where re-use of plastic components from the e-wastes is not possible, mechanical recycling is opted. In mechanical shredding and sorting processes for e-scrap (pre-treatment), large volumes of various mixed plastic fractions are usually generated. Often, it is not economically viable to separate these further and recover the various types of plastic. There is also a difficulty of achieving acceptable product quality. Limited markets for recycled polymers and innovations in polymer performance and consumer acceptance add to the problems. This is because of the contamination during sorting process of plastics and subsequently during the recycling process affect the product quality adversely. Furthermore, all residual metals still contained in them would inevitably be lost. Separating residual metals by mechanical means can become quite costly while generating only little additional metal value. Mechanical recycling of older WEEE which contains plastics cause problems when dismantling processes do not utilize adequate means, facilities and trained people. 

Plastic content in the e-waste is used either as a chemical feedstock to replace the reducing agents, CO and H_2_, or as fuel [96] due to their high calorific values. Without strict temperature control during extrusion, there is a potential risk of generating dioxins and furans from some halogenated flame-retardants [96], which can be addressed as already discussed.

### 4.6. Carbon Sequestration

#### 4.6.1. Refractories (Magnesium Production, Magnesia-Carbon Production)

Global growth in the magnesium market is expected to reach 1.2 Mt/year by 2020 (Magnesium Metal, 2016). Application of Mg has been limited by relatively high cost of production, associated energy costs and a high CO_2_ emission. The main environmental concerns to produce different types of MgO are the use of energy and emissions to air. The calcination process of magnesite has a high carbon footprint. It is the principal user of energy and the main source of emissions, with 1.1 t CO_2_ resulting from the carbonate decomposition to produce 1 t of magnesia (MgO) accounting for direct CO_2_ emissions of 6.7 Mt from the calcination process alone. A study reports use of virgin epoxy resin and biomass can reduce 99% of the carbon dioxide produced in thermal processing of magnesium carbonate while simultaneously producing hydrogen [9]. 

Magnesium metal production (where the intermediate product can be magnesia as described in the research) is another high carbon footprint reaction. Use of plastics can greatly reduce the emissions, energy and process temperatures not only during the intermediate MgO formation, but also during the carbothermic reduction to magnesium metal, eliminating the need for the non-renewable coke. The above study [9] suggests a potential decrease in global warming potential up to 99% for carbothermic reduction to elemental magnesium from MgO or MgCO_3_, dead burnt MgO production and magnesia carbon refractory bricks production in presence of plastics and biomass with synergistic effects. The current operating temperature of ≈1600 °C can be brought down by at least 100–200 °C due to the reducing gases produced by the plastics and the biomass enhancing the energy efficiency of the process [99]. Magnesium carbonate can also adequately control the toxin emissions during the pyrolysis of plastics and the biomass. Controlling the ratios of MgCO_3_:biomass:epoxy resulted in H_2_ enriched product paving way to cleaner hydrogen economy.

#### 4.6.2. Clinker Production

Cement production accounts for about 22% (1.5 Gt in 2005) of the industry sector’s total direct CO_2_ emissions. Two thirds of this (0.94 Gt per year in 2005) is generated by the decomposition of limestone into cement clinker and CO_2_. The remaining one third is from fuel combustion. The total CO_2_ emissions during the cement making process are mainly due to:raw material decomposition (59%),fuel combustion (26%), andelectricity consumption (12%).

The overall CO_2_ emission factor is 0.81 t/tonne cement. Producing one tonne of clinker will have: 1 t × 65% × 0.79 = 0.51 t of CO_2_ emitted from CaCO_3_ decomposition. The magnesium and calcium have similar chemistry and a similar reduction in emissions, global warming potentials, energy and process temperature can be achieved in Clinker production. Use of plastics and biomass can reduce the emissions during raw material decomposition in addition to energy recovery.

## 5. Distinct Advantages and Limitations of Using Polymers in Mineral and Metallurgical Processing

Plastics, used or virgin, have a higher calorific value than coal and can provide a much-needed local energy supply [2]. Pyrolysis of waste thermosets and thermoplastics generate CH_4_, H_2_, CO, CO_2_ and other hydrocarbons used as resources in high temperature metallurgical processing, e.g., in blast furnace and EAFs. In Japan, mechanical recycle, feedstock recycle, and energy recovery were used to increase the waste plastic utilization over 50 wt. % ratio to solve the problem of shortages in landfills and incineration sites. Consumable carbon anodes for the aluminium industry using high temperature interactions of waste plastics with petroleum coke have been developed [100].

The high temperature processability of the thermoplastics finds applications blast furnaces, especially iron and steel processing [55,56]. Waste plastics are thermally decomposed into approximately 20 wt. % coke, 40 wt. % hydrocarbon oil (tar and light oil) and 40 wt. % coke oven gas, which are utilized as iron ore-reducing agents in blast furnaces, raw materials for the chemical industry and a fuel for power plants, respectively. There is a higher carbon uptake when plastics are used compared to the coke in reduced iron. An added advantage of waste plastics utility in blast furnace operations is their low sulphur and alkali content. Plastic production uses far less energy compared to traditional materials.

### 5.1. Mitigate GHG and Toxic Emissions

At the outset, injecting waste plastics, which have a higher calorific values (CV) than coal, decreases blast furnace coke and energy consumption and lowers CO_2_ emissions by about 30% in comparison to coke and coal depending on the type of polymers used as shown in Figure 18 [25,59], according to the Reactions (5) and (6):Fe_2_O_3_ + 3CO → 2Fe + 3CO_2_ (coke, pulverized coal)(5)
Fe_2_O_3_ + 2CO + H_2_ → 2Fe + 2CO_2_ +H_2_O (using plastic)(6)

Reaction (5) occurs at temperatures > 700 °C; Reaction 6 occurs at temperatures < 580 °C [101,102].

PET has negative CO_2_ emission reduction potentials, i.e., there is an increase of CO_2_ emission (Figure 18). This is attributable to their relatively small calorific values and carbon (coke product yield) and hydrogen contents, which leads to a relatively small coke substitution effect when using these plastic resins. Overall, the net emissions for pyrolysis of plastics are –160 kgCO_2_ [103,104] (Table 13). Bio-fuel delivered during pyrolysis can replace fossil fuel with no toxic or harmful emissions. The dioxins generation is contained in pyrolysis as opposed to incineration. Waste plastics also reduces SO_2_ emissions due to a low or nil sulphur content in plastics. Magnesium carbonate can also adequately control the toxin emissions during the pyrolysis of plastics and the biomass. Carbon sequestration in a high carbon footprint reaction using virgin thermosets has resulted in 99% CO_2_ reduction [9].

### 5.2. Replaces or Minimises Required Resources and Non-Renewables (Dematerialization)

One kilogram of waste plastics replaces ≈0.75 kg of coke, 1.3 kg of coal or 1 kg of heavy oil [56] in EAF steelmaking. A partial replacement of conventional metallurgical coke with high density polythene (HDPE) waste produces significant levels of slag foaming and lowers the energy consumption by 8–15 kWhr/tonne [105]. 

The virgin epoxy resin used in palletisation serves as the reductant, minimising the need for non-renewable resources, such as coke, as the reductant and fuel. The conventional iron ore reduction/palletization requires at least 1:3 Fe_2_O_3_ to C and a binder. When epoxy resin is used as the binder it acts as reductant and fuel at ≈1:0.3 Fe_2_O_3_ to C, i.e., at least 10 times less carbonaceous material is used. The pyrolysis of resins provides for reducing atmosphere of H_2_ and CO from the methane generated at the reaction temperatures, which aids the reduction reaction [7]. 

### 5.3. Reduction in Process Temperatures and Energy

Reduction of manganese ore in the solid state at temperatures below 1100–1200 °C by solid carbon, hydrogen or carbon monoxide does not go beyond MnO. Using methane-containing gas, manganese oxides can be reduced to manganese carbides at relatively low temperatures when manganese ore is solid. Reduction of MnO by carbon requires a higher equilibrium temperature (1340 °C) compared to MnO reduction by methane (928 °C), as shown in Reactions (7) and (8) [106,107]:MnO + 10/7 C → 1/7 Mn_7_C_3_ + CO(7)

The Gibbs free energy change, ΔG° for the above reaction is 257.75−0.1598T J·mol^−1^ [107].
MnO +10/7 CH4 = 1/7 Mn_7_C_3_ + CO + 20/7 H_2_(8)

ΔG° = 377.68−0.3144T Jmol^−1^.

The exponential increase in the increasing values of equilibrium constants for Reaction (8) is 8.46 at 1000 °C, 113.80 at 1100 °C and 1075 at 1200 °C (Figure 19). The increasing carbon activity in the CH_4_-H_2_ gas relative to graphite increases with temperature. This makes the reduction of manganese ore by methane-containing gas faster than in the conventional carbothermal reduction process [106]. 

The pellets using epoxy resin as the binder show high compression strength of 4 kN without heat treatment compared to the industry requirement (2.5 kN) of an indurated pellet processed at 1200–1350 °C. Energy consumption during heat induration of magnetite concentrates ranges from 320,000 kJ/t to 740,000 kJ/t pellets and higher for higher non-magnetite ores due to the absence of exothermic and oxidation reactions. Cold bonded pellets being a low temperature alternative to heat-indurated pellets consume less energy for bonding self-reducing pellets and for refractory ores that are difficult to heat treat. The estimated cold bonding capital and operating costs is 2/3 of that of heat induration [7]. 

The FeSi and SiC formation would normally occur at >1800 °C in conventional process. Use of virgin silicones and epoxy resins has seen FeSi and SiC formations around 1500 °C [8]. The saved energy resources add up to 60 GJ/t as the result of plastic replacement in blast furnaces [4]. In EAF steelmaking, replacing ≈30 % of the coke and coal by polyethylene wastes resulted in energy savings up to ≈12 kWh/t of plastic charge [64]. Tyre injection trials at OneSteel’s Sydney steel mill, Australia, saw reduction in electrical energy consumption from 1.526 MJ/kg to 1.483 MJ/kg billet (0.043 MJ/kg billet savings in energy), the amount of carbon injectant reduced from 464 kg/heat to 406 kg/heat, and improved number of liquid tonnes per power-on time minute from 2.12 to 2.2 t/min [64]. 

## 6. Disadvantages

Chlorinated polymers like PVC may lead to dioxin formation during landfilling and incineration. Exposure to dioxins can cause skin lesions, damage the immune system, interfere with hormones and also can cause cancer. However, dioxins are contained within the required level during pyrolysis of waste plastics. Heavy metals emissions, especially the Cd and Hg from the blast furnaces, could pose problems when the plastic wastes are sourced from mechanically separated packaging and commercial wastes and the lighter fractions from the end of life vehicles [108]. However, the input loads of heavy metals loads can efficiently be controlled using filter and cleaning devices.

## 7. Conclusions

Concerns are mounting regarding the usage, disposal and accumulation of EOL plastics in landfills and in natural habitats. Ingestion or entanglement in plastic and leaching of chemicals from plastic products can affect wild and marine life and humans. Currently there is a global consensus that plastics are a threat to the environment prompting many governments to ban single use plastic products, including Australia, India, Kenya, U.K., Taiwan, Zimbabwe, Montreal (Canada), Malibu and Seattle (USA), Hamburg (Germany), France, Morocco, and Rwanda. About 4% of world oil production is used as a feedstock to make plastics and a similar amount is used as energy in the process. When one third of the plastics produced are used in packaging and other short-lived applications only to be rapidly discarded [109], such use of plastics is unsustainable or untenable. Solving the pollution problems of plastics will require appropriate planning and design for end-of-life recyclability and increased recycling capacity through the combined actions of the public, industry, scientists and policymakers. This review highlights many current as well as novel future applications of recycled and virgin polymers in metallurgical and mineral processing which could greatly contribute in managing EOL plastics. Specific applications include: Coal dewatering, where super absorbent polymers offer ambient non-evaporative technology to dry low-grade brown coals and fine black coals saving on costs, energy and emissions while simultaneously enhancing the calorific values of the same. These super absorbent polymers can be regenerated at low pH or low temperatures (37–52 °C)Chemical feedstock recycling of waste plastics in coke oven and blast furnaces produce carbon residues and reducing gases such as methane and hydrogen and fuel oils. They reduce the coke consumption in blast furnace iron making and CO_2_ emissions >30%. In EAF, they aid slag foaming contributing to energy reductions. Harmful emissions of dioxins are reduced during the feedstock recycling of waste plastics.For iron ore palletisation in this novel process, the epoxy resins used as the binder, reductant and as fuel, offer high compression strength under ambient conditions, far exceeding the heat induration strength, saving on costs, energy and emissions. In addition, the reducing atmosphere provided by the polymer resins reduce the reaction temperature by about 100 °C. Use of the bio-renewable epoxy resins, currently being commercialised, makes this a green process.Iron alloy production, where silicones and the epoxy resins are used as the alloying agents, binder, reductant and fuel towards the production of ferrosilicon, silicon carbide with varying microstructures. The alloy formation occurs at 1600 °C as opposed to the conventional 2000 °C.The polymer resins reduce the non-renewable coke consumption in the above processes.Carbon sequestration, where the plastics used totally consume the CO_2_ produced in a high carbon high footprint reaction, such as thermal decomposition of magnesite or clinker production, simultaneously producing hydrogen, contributing to hydrogen economy.The plastics in e-wastes are used as fuel and reductant in recovering valuable metals.Recycling of used polymers offers competent waste management strategy reducing the land-fill burden.

## Figures and Tables

**Figure 1 materials-12-00655-f001:**
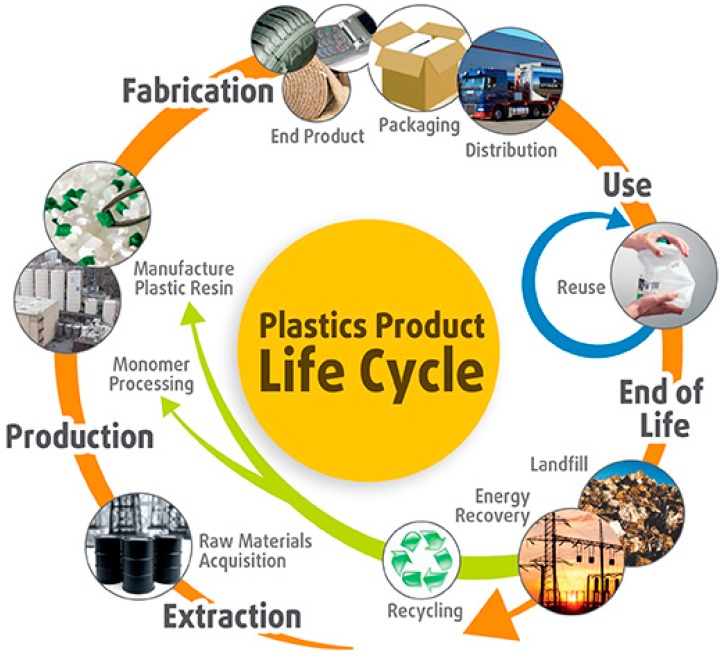
Life cycle of plastics [2].

**Figure 2 materials-12-00655-f002:**
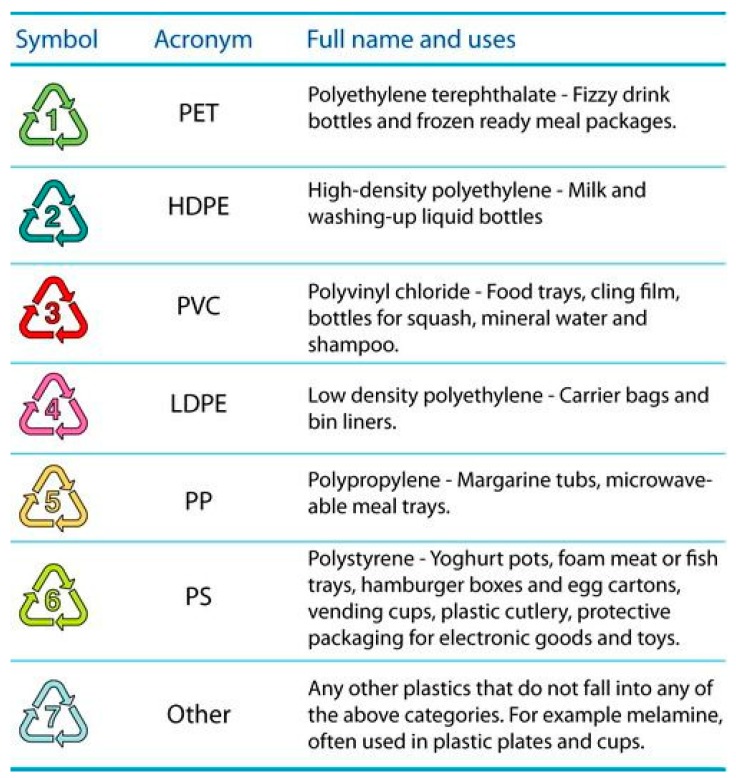
Plastics identification code. Adapted from [6], with permission from © 2016 Elsevier.

**Figure 3 materials-12-00655-f003:**
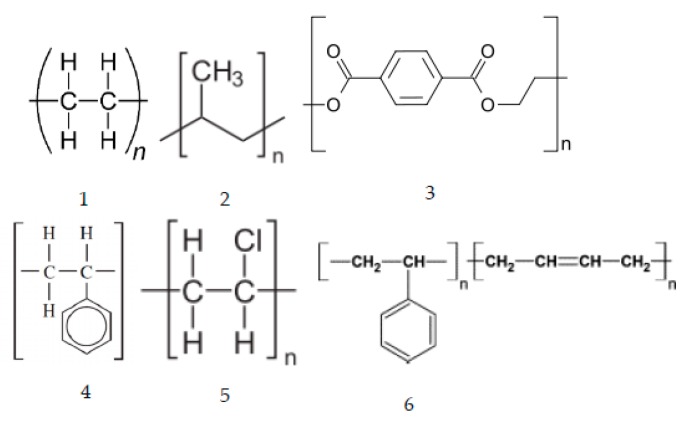
Thermoplastics used in iron ore reduction: (**1**) polyethylene, (**2**) polypropylene, (**3**) polyethylene terephthalate, (**4**) polystyrene, (**5**) polyvinyl chloride and (**6**) styrene-butadiene rubber (SBR).

**Figure 4 materials-12-00655-f004:**
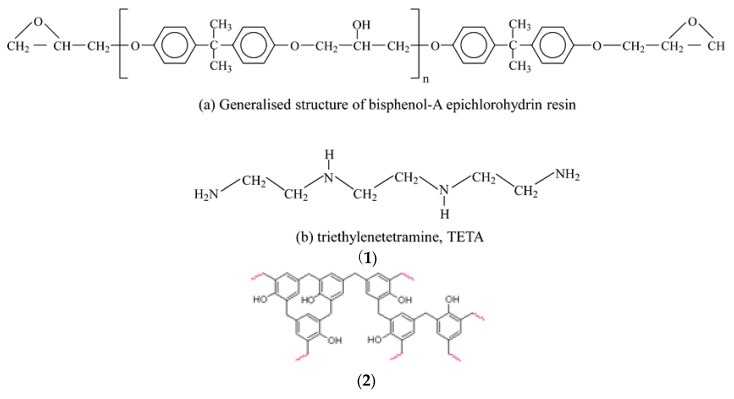
Structures of thermosets: (**1**) epoxy resin, a. resin, b. hardener; (**2**) cross-linked Bakelite.

**Figure 5 materials-12-00655-f005:**
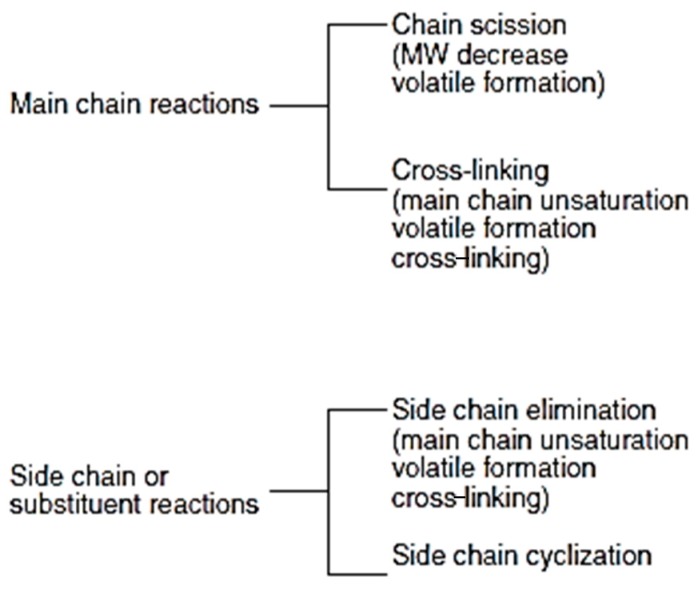
Thermal decomposition mechanisms in polymers [20].

**Figure 6 materials-12-00655-f006:**
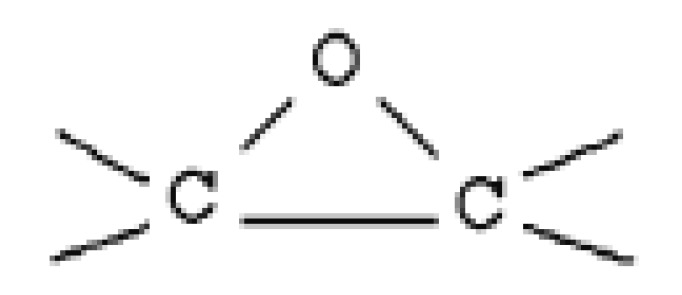
Three-member epoxy ring.

**Figure 7 materials-12-00655-f007:**
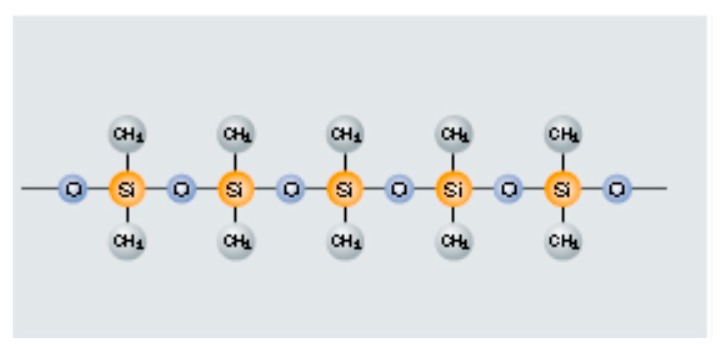
Linear silicone polymer (polydimethylsiloxane).

**Figure 8 materials-12-00655-f008:**
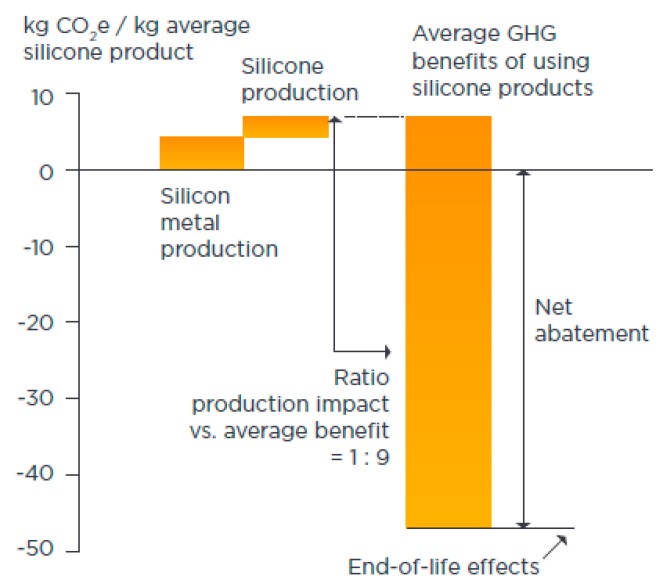
GHG benefits of using silicones [37].

**Figure 9 materials-12-00655-f009:**
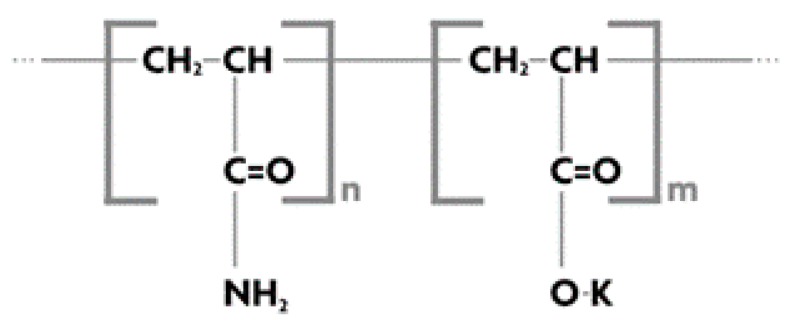
Structure of Aquasorb.

**Figure 10 materials-12-00655-f010:**
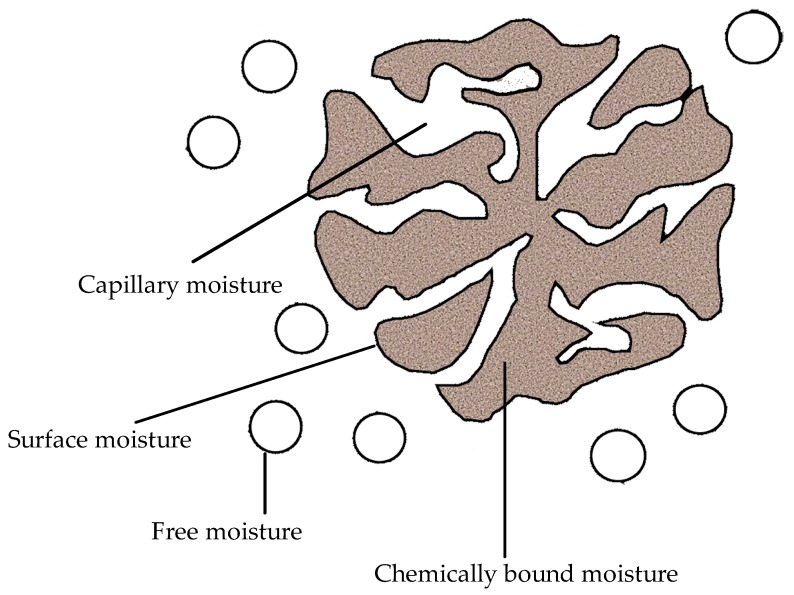
Schematic representation of different types of moisture distribution within coal fines [44].

**Figure 11 materials-12-00655-f011:**
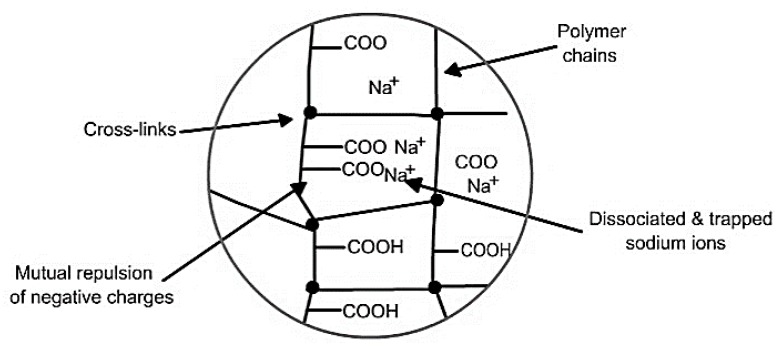
Mechanism of water absorption in super-absorbent polymers (SAP).

**Figure 12 materials-12-00655-f012:**
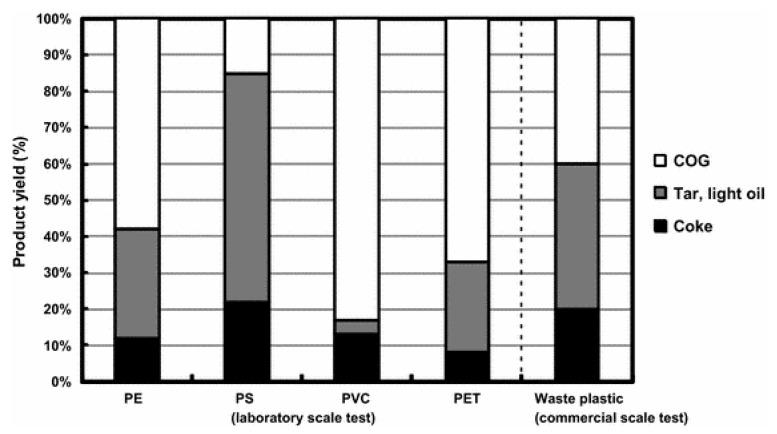
Effect of different plastic additions on the decomposition product yield (mass %) (Adapted from [50], with permission from © 2015 Springer Nature).

**Figure 13 materials-12-00655-f013:**
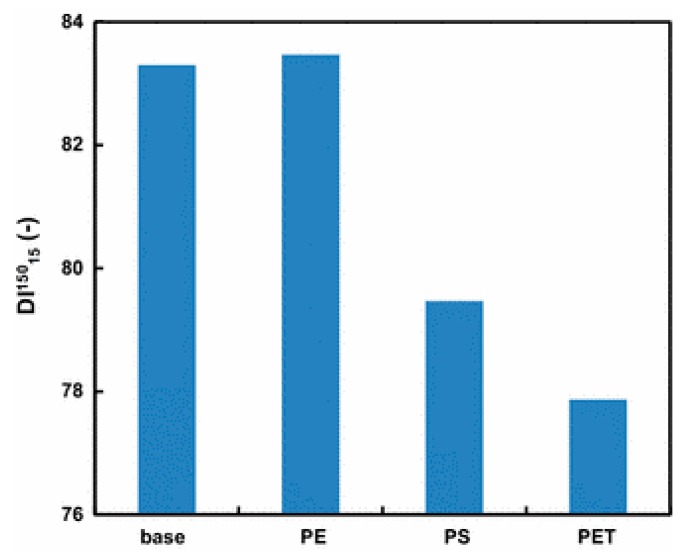
Effect of adding various plastics (2% mass) on coke strength $DI_{15}^{150}$ (Adapted from [50], with permission from © 2015 Springer Nature).

**Figure 14 materials-12-00655-f014:**
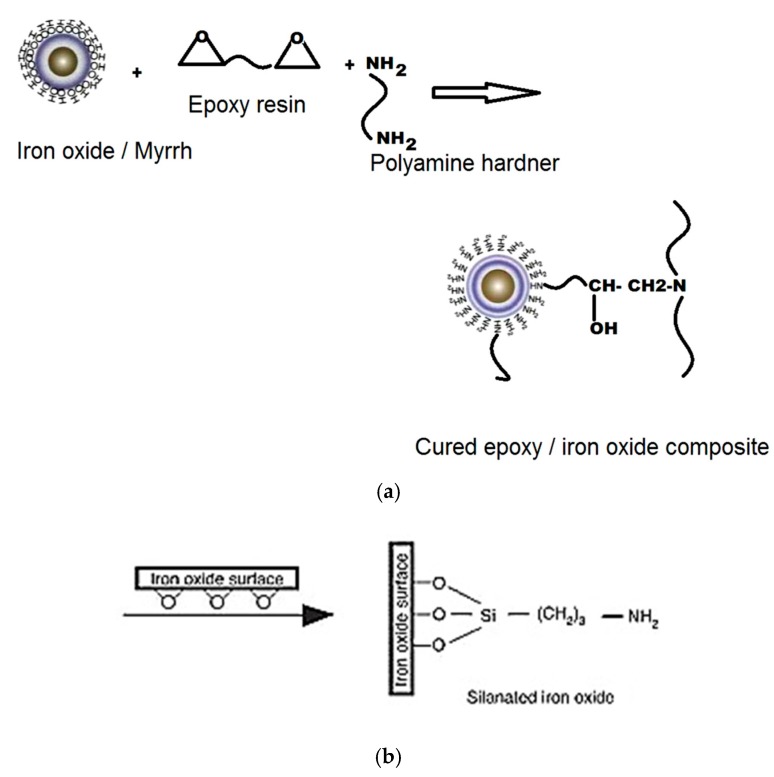
(**a**) Interfacial chemical bonding between iron oxide and the epoxy resin. Numbers indicate bond energies (kJ·mol^−1^) [29]. (**b**) Iron oxide and silicone [83].

**Figure 15 materials-12-00655-f015:**
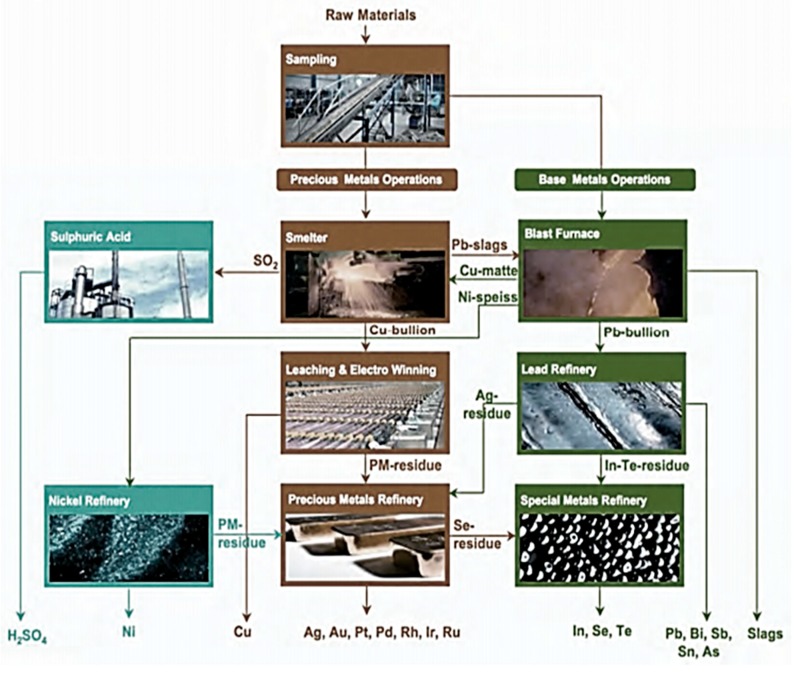
Umicore Integrated smelting refining operations.

**Figure 16 materials-12-00655-f016:**
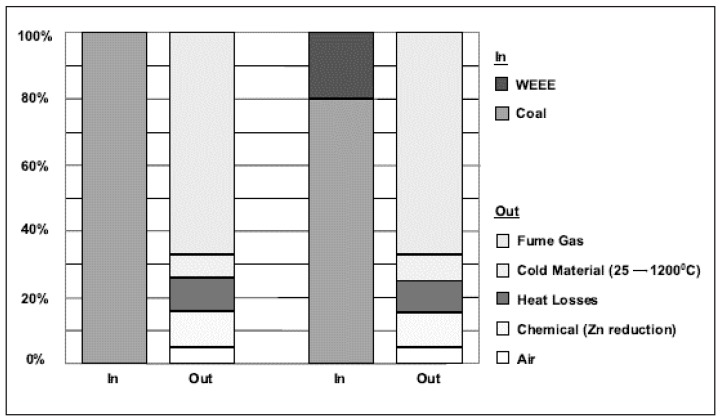
Influence of plastics on the energy balance in the furnace [96].

**Figure 17 materials-12-00655-f017:**
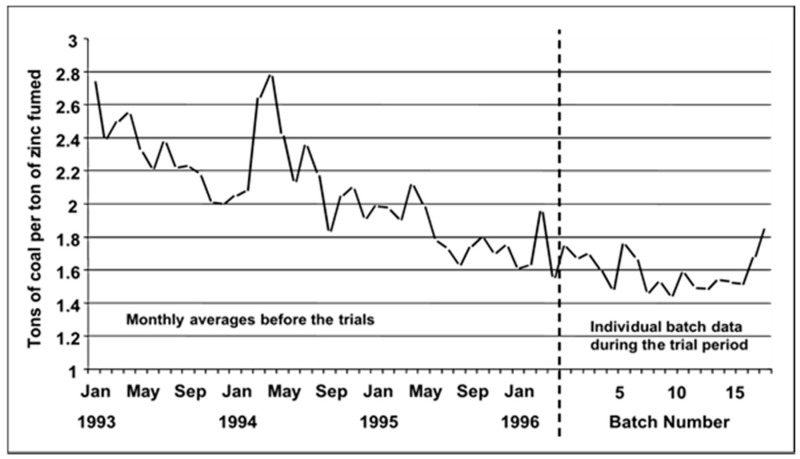
Monthly average coal consumption rate [96].

**Figure 18 materials-12-00655-f018:**
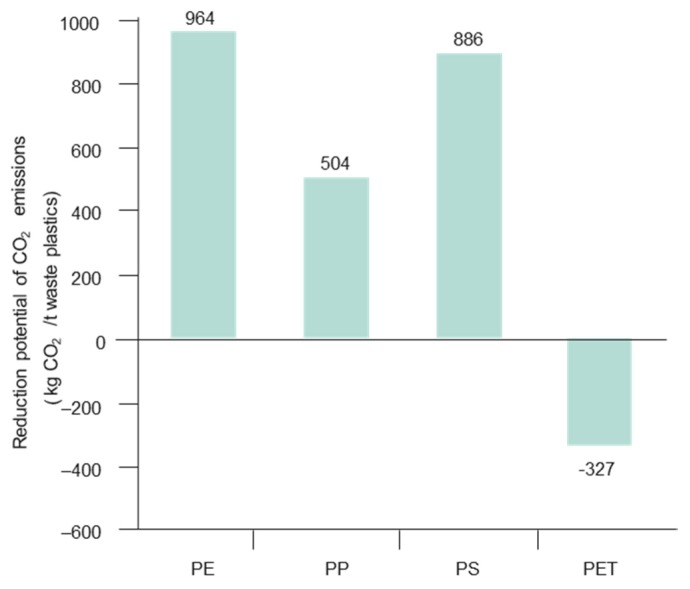
Reduction potential of CO_2_ emissions with 1 t waste plastics injection, Adapted from [25], with permission from © 2009 Springer Nature.

**Figure 19 materials-12-00655-f019:**
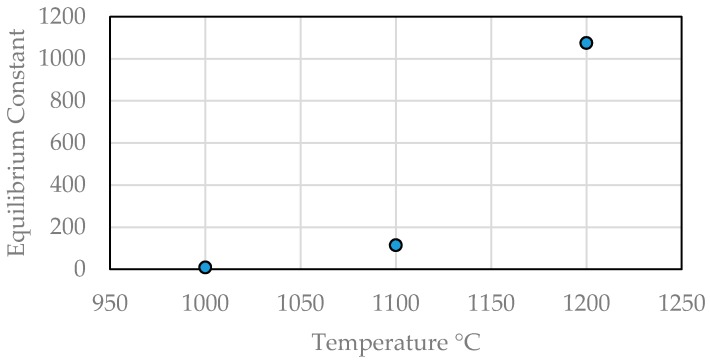
Effect of temperature on the equilibrium constant of the reduction reaction of MnO to Mn_7_C_3_.

**Table 1 materials-12-00655-t001:** Global primary plastics production, lifetime distribution of the product and primary waste generation (in million metric tons) in 2015 according to industrial use sector [3].

Market Sector	2015 Primary Production (Mt)	Mean (yrs) Life	2015 Primary Waste Generation (Mt)
Packaging	146	0.5	141
Transportation	27	13	17
Building and Construction	65	35	13
Electrical/Electronic	18	8	13
Consumer & Institutional Products	42	3	37
Industrial Machinery	3	20	1
Textiles	59	5	42
Other	47	5	38
Total	407	-	302

**Table 2 materials-12-00655-t002:** Share of polymer types produced according to industrial sector (2002–2015) [3].

* Polymer Type/Additive	2015 Primary Production (Mt)	2015 Primary Waste Generation (Mt)	Share of Polymer Types Produced during 2002–2014 according to Industry Sectors								
			TRANSPORTATION %	Packaging %	Building and Construction %	Electrical/Electronic %	Consumer & Institutional Products %	Industrial Machinery %	Textiles %	Other %	Total %
LDPE	64	57	0.1	13.5	1.1	0.5	2.9	0.2		1.7	20.0
HDPE	52	40	0.8	9.3	3.3	0.2	1.7	0.1		0.9	16.3
PP	68	55	2.6	8.2	1.2	0.9	3.8	0.2		4.2	21.0
PS	25	17	0.0	2.3	2.2	0.6	1.8	0.0		0.7	7.6
PVC	38	15	0.3	0.9	8.1	0.4	0.6	0.0		1.4	11.8
PET	33	32	0.0	10.1	0.0	0.0	0.0	0.0		0.0	10.2
PUR	27	16	1.6	0.2	2.4	0.4	1.0	0.3		2.5	8.2
Other	59	42	1.4	0.1	0.5	1.0	0.2	0.0		1.7	4.9
PP&A fibers	16	11	-	-	-	-	-	-	100		100
Additives	25	17	-	-	-	-	-	-	-	-	-
Total (non-fibres only (2002–2014)	-	-	6.7	44.8	18.8	3.8	11.9	0.8	-	13.2	100.0
Total (2015)	407	302	-	-	-	-	-	-	-	-	-

* LDPE (low-density polyethylene); HDPE (high-density polyethylene); PP (polypropylene); PS (polystyrene); PVC (polyvinyl chloride); PET (polyethylene terephthalate); PUR (polyurethane); PP&A fibres (polyester, polyamide, and acrylic fibres).

**Table 3 materials-12-00655-t003:** Yield in thermal and catalytic pyrolysis of HDPE and LDPE [22].

Pyrolysis Product Yield (wt. %)		HDPE (450 °C)	HDPE (550 °C)	LDPE (550 °C)
Gas Fraction		13.0	16.3	14.6
Liquid Fraction	Total	84.0	84.7	93.1
	C6–C12	56.55		
	C13–C23	37.79		
	>C23	5.66		
Solid Fraction		3.0	-	-

**Table 4 materials-12-00655-t004:** Offgas generated from different plastic resins in blast furnace [25].

Offgas	Without Plastics	PE	PP	PS	PET
Total blast furnace gases (BFG) m^3^/thm	1670.9	1747.3	1787.2	1684.4	1741.8
CO vol %	22.9	21.1	21.2	21.9	22.4
CO_2_ vol %	21.3	20.2	20	20.8	20.6
H_2_ vol %	4.6	7.3	7.2	6.3	5.4
H_2_O vol %	2.4	3.8	3.8	3.3	2.7
N_2_ vol %	48.9	47.5	47.9	47.8	48.9
CV of BFG MJ/m^3^	3.4	3.47	3.46	3.44	3.41

**Table 5 materials-12-00655-t005:** Properties of resins [25,27,28].

Properties	Coke	PE	PP	PS	PET	PVC	Bakelite	Epoxy
Calorific value (kJ/kg)	25,000–30,000	44,800	42,700	41,900	23,200	1800	NA	≈32,000
Carbon contents (kg C/kg)	-	0.86	0.86	0.92	0.62	0.38	0.53	0.70

**Table 6 materials-12-00655-t006:** Product yields from the pyrolysis of composites [26].

Sample Description	Temperature (°C)	Solids (wt. %)	Oil/Wax (wt. %)	Gas (wt. %)	Oil/Gas Ratio
PE1: Thermoset polyester/styrene resin with calcium carbonate and aluminium trihydrate fillers (36 wt. %), glass fibre reinforcement (12 wt. %)	350	82.9	14.5	2.6	5.6
	400	52.6	41.2	6.2	6.7
	450	48.7	45.0	6.3	7.1
	500	45.8	45.7	8.5	5.4
	650	46.6	47.0	6.4	7.3
	800	38.2	47.4	14.4	3.3
PH1: Phenolic resin with magnesium oxide and calcium carbonate fillers (41 wt. %), glass fibre reinforcement (31 wt. %)	400	94.3	5.1	0.6	8.5
	500	90.2	8.8	1.0	8.8
	600	86.6	10.9	2.6	4.2
	700	85.3	12.0	2.8	4.3
	800	83.4	11.9	4.6	2.6
EP: Epoxy resin, inorganic fillers (30 wt. %) glass and carbon fibre reinforcement (45 wt. %)	350	81.7	18.0	0.2	90
	400	70.7	27.6	1.7	16.2
	500	67.4	31.3	1.2	26.1
	600	69.6	29.4	1.0	29.4
	800	65.3	31.7	3.0	10.6
PE2: Thermoset polyester/styrene resin, gel-coat surface, glass fibre reinforcement (30 wt. %)	450	32.6	64.1	3.3	19.4
PH2: Phenolic resin, Nomex paper core, carbon fibre reinforcement	550	65.0	30.4	4.6	6.6
PP: Polypropylene resin, glass fibre tape (75 wt. %)	550	78.9	20.0	1.1	18.2
VE: Vinylester resin with woven glass fibre fabric (70 wt. %)	550	83.4	15.0	1.6	9.4

**Table 7 materials-12-00655-t007:** Total yields of gases derived from the pyrolysis of EP in relation to final pyrolysis temperature. Adapted from [26], with permission from © 2003 Elsevier.

Gas Yield (mol·g^−1^ × 10^−4^)	Final Pyrolysis Temperature (°C)				
	350	400	500	600	800
Carbon dioxide	2.31	7.20	4.55	13.68	26.19
Carbon monoxide	0.00	0.44	1.13	0.55	4.04
Hydrogen	0.00	0.00	0.02	0.08	0.63
Methane	0.12	3.74	1.10	1.04	2.73
Ethane and ethene	0.15	0.42	0.84	0.42	1.22
Propane	0.05	0.16	0.21	0.08	0.25
Propene	0.09	0.27	0.27	0.08	0.37
Butane	0.00	0.01	0.01	0.01	0.01
Butene	0.01	0.00	0.01	0.00	0.01
Butadiene	0.00	0.00	0.01	0.00	0.03
GCV (MJ/m^3^)	51.1	39.8	42.0	28.9	23.9

**Table 8 materials-12-00655-t008:** Bond strengths at the interface [35].

Bond	Bond Energy kJ·mol^−1^
Ionic	600–1000
Covalent	60–700
Metallic	100–350
Strong H bonds (involving F)	Up to 40
Other H bonds	10–25
Dipole-dipole	4–20
Dipole-induced dipole	1–2
London dispersion forces	0.1–40

**Table 9 materials-12-00655-t009:** Water absorption of different materials.

Material	Water Absorption wt. %
Whatman No. 3 filter paper	180
Facial tissue paper	400
Soft polyurethane sponge	1050
Wood pulp fluff	1200
Cotton ball	1890
SAP, i.e., Superab A-200	20,200

**Table 10 materials-12-00655-t010:** Product yields of plastics in coke oven (%) at ≈1010 °C [25].

	PE	PS	PET	PVC	PP
Gas %	58 ^a^	15 ^a^	67 ^a^	83 ^a^	40 ^b^
Oil %	29 ^a^	63 ^a^	25 ^a^	4.0 ^a^	44 ^b^
Coke %	13 ^a^	22 ^a^	8.0 ^a^	14 ^a^	16 ^b^
Carbon content (kg C/kg)	0.86 ^b^	0.92 ^b^	0.62 ^b^	0.38 ^b^	0.86 ^b^
Volatile matter %	17 ^b^	3.4 ^b^	39 ^b^	25 ^b^	15 ^b^

^a^ Experimental value (measured using a laboratory-scale coke ovens). ^b^ Calculated value.

**Table 11 materials-12-00655-t011:** Material Composition of electronic equipment. Adapted from [93], with permission from © 2015 Elsevier.

Products	I	II	III	IV	V	VI	VII	VIII	IX	X	XI	XII	XIII	XIV
Materials							g/unit							
Aluminum	0.77		67			242	130	130	12	2.9	1370	441	441	
Antimony	0.001	0.001	14	0.71	0.71					0.084				0.154
Arsenic	2.5	2.5												0.002
Barium						1								0.49
Beryllium										0.003				
Cadmium			0.2								0.407			
Cerium	≤0.001	≤0.001		0.005	≤0.001		≤0.001	≤0.001						≤0.001
Chromium	0.07	0.07	0.03											
Cobalt	0.065	0.065												
Copper	135	135	656	824	824									
Dysposium	0.06	0.06												
Europium	≤0.001	≤0.001		0.008	≤0.001									
Ferrite						483								
Gadolinium	≤0.001	≤0.001		≤0.001	0.002		≤0.001	0.002						≤0.001
Gallium		0.016			0.005		0.003	0.003			0.119			
Glass			15,760	162	216	6845	590	590		10.6	6915			
Gold	0.22	0.22		0.11	0.11	0.31	0.2	0.2	0.024	0.038		0.005	0.005	0.044
Indium	0.04	0.04		0.003	0.003		0.079	0.082			0.012			0.008
Lanthanum	≤0.001			0.007			≤0.001							≤0.001
Lead	5.3	5.3	1319			464	16		1	0.6				1.1
Mercury	≤0.001	≤0.001					≤0.001	0.004	1					≤0.001
Molybdenum	0.04	0.04					0.633	0.633			0.295			0.008
Neodyumium	2.1	2.1								0.005		1		0.427
Nickel	3.6	3.6				199			1	1.5				0.722
Palladium	0.04	0.04		0.044	0.044		0.04	0.04	0.009	0.015		0.003	0.003	0.008
Plastics			8755	612	573	2481	1780	1780	63	60	1172	44	44	
Platinum	0.004	0.004								0.004				
Praseodymium	0.274	0.274		≤0.001			≤0.001			0.001		0.145		0.055
Selenium											0.119			
Silicon									5		226			
Silver	0.25	0.25		0.45	0.45	1.25	0.52	0.52	1	0.244		0.031	0.031	0.05
Steel/Iron			2088			3322	2530	2530	11	8		62	62	
Tantalum	1.7	1.7												
Tellurium											0.406			
Terbium	≤0.001			0.002			≤0.001							≤0.001
Tin			32	18	18	20	24	24	1	1	0.116			
Titanium							0.633	0.633						
Tungsten							0.633	0.633						
Vanadium						1								
Yttrium	0.002	0.002		0.11	0.005	1	0.016	≤0.001						≤0.001
Zinc	0.004	0.004	8.6						4	1	0.4			≤0.001
# of critical raw materials	14	13	1	10	8	1	10	7	2	8	2	4	1	14
# of precious metals	4	4	0	3	3	2	3	3	3	4	0	3	3	3

I = LCD Notebooks; II = LED Notebooks; III = CRT TVs; IV = LCD TVs; V = LED TVs; VI = CRT Monitors; VII = LCD Monitors; VIII = LED Monitors; IX = Cell Phones; X = Smart phones; XI = PV Panels; XII = HDDs; XIII = SSDs; XIV = Tablets.

**Table 12 materials-12-00655-t012:** Dioxin and furan mass balance for the zinc fuming plant [96].

**Input**						
Feed	Quantity (tons)		PXDD/F Concentration (mg/kg)			Flux (g/batch)
Slag	86		0			0.0
Steel Dust	5		30.6			0.15
Cold Slag	5		0			0.0
PC Scrap	5		1250			6.25
Total Input						6.4
**Output**						
Output	Quantity		PXDD/F Concentration			Flux (g/batch)
	(m^3^/h)	(tons)	(µg/kg)	(ng/m^3^)	Sampling Time (h)	
Stack Gas	130,000			182	2	0.047
Raw Fume		8.48	6.68			0.057
Total output						0.104

**Table 13 materials-12-00655-t013:** GHG emissions for different disposal options [103].

Heading	GHG Emissions (kg CO_2_ (e) Per Tonne of Mixed Plastic)				
	Input Materials	Transport	Processing	Displacement Savings *	Net Emissions
Landfill	0.0	15.1	55.7	0.0	70.8
Incineration	0.0	15.1	2408.0	−565.5	1857.6
Pyrolysis	13.0	197.2	55.6	−425.5	−159.7
Gasification w/MTG (methanol-to-gasoline process)	153.7	153.7	995.5	−261.7	1041.2
Gasification w/F-T (Fischer–Tropsch process )	153.7	139.3	285.2	−147.1	431.1
Gasification w/Bio (gasification with biological conversion of syngas to ethanol)	153.7	187.7	1217.1	−454.9	1103.6
Catalytic depolymerisation	16	197.5	51.0	−397.4	−132.8

* The avoided greenhouse gas (GHG) emissions associated with the displacement, where reuse occurs, and other product manufacture is displaced.

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
