# Peer review of "Plastics—Villain or Hero? Polymers and Recycled Polymers in Mineral and Metallurgical Processing—A Review"

_materials, 2019, doi:10.3390/ma12040655_

Reviewer 1 Report

This is an interesting article on plastics.  Article is well written. Following minor corrections are needed in the article

-Introduction should be re-written. At the moment it is too precise. Authors should also some light on importance of bio-plastics

-Figure 3 should be reproduced. The text of the figures is not visible

-Relevant articles on recycling should be cited such as : Sustainability of bioplastics: Opportunities and challenges, 13,  68-75 (2018) ;  Recent developments in recycling of polystyrene based plastics, 13,  32-38 (2018) etc

Author Response

Response: Thank you for the positive comments. The time and the valuable comments of the reviewers much appreciated.

Point 1. Introduction should be re-written. At the moment it is too precise. Authors should also some light on importance of bio-plastics

Response 1. Scientific articles characterised by precision is a desirable feature. Please check my response to Point 3 below. The submitted paper discusses only about the polymers relevant to mineral and metallurgical processing. At the moment the applications discussed in the paper for recycled polymers are mainly based on thermal degradation. There is, however, a mention of Biorenewable epoxy resins having potential future application, in reference 83, (G. Yang, B. J. Rohde, H. Tesefay and M. L. Robertson, “Biorenewable epoxy resins derived from plant-based phenolic acids,” ACS Sustainable Chem. Eng., vol. 4, no. 12, p. 6524–6533, 2016).

Point 2. Figure 3 should be reproduced. The text of the figures is not visible

Response 2: Figure 3 is replaced with a better version.

Point 3. Relevant articles on recycling should be cited such as: Sustainability of bioplastics: Opportunities and challenges, 13, 68-75 (2018); Recent developments in recycling of polystyrene based plastics, 13, 32-38 (2018) etc

Response 3. We respectfully disagree. This article is a specialised article and not about the recycling of polymers, but application and utilization of the plastics in mineral and metallurgical processing. The article discusses only about the plastics that are relevant to the specific industry. As such many plastics such as polystyrene and bioplastics are not included as their applications in the industry is not fully explored. The above references do not shed light on their applications in mineral and metallurgical processing.

Reviewer 2 Report

This is an interesting and valuable review including many references to literature data. Use of different plastics in metallurgical processes reduces CO2 emission by ~30 %, in comparison to coke and coal. Thermoplastic polymers may serve as a partial substitute for coke in iron steel industries and are more profitable than thermoset resins. This is an ecological alternative for classical recycling of plastics.

Author Response

Response: Thank you for the positive comments. The time and the valuable comments of the reviewers much appreciated

Reviewer 3 Report

The present work is a review article dealing with the use of polymers in mineral and metallurgical processing. In particular, the authors discussed the application of both recycled and virgin plastic for various aims, as the coal treatment, the iron production, the ore palletisation, the iron alloy production, the manganese processing and other, focusing the attention on the advantages and the disadvantages. The topic of this work is quite interesting, there are few revisions to be incorporated in the text before publication:

The quality of figure 3 is too low: it is impossible to read its content! The authors are invited to fix that issue.

Paragraphs 3.4 and 4.1 should be combined together in a single paragraph.

Authors are invited to check and standardize text styles throughout the paper.

Author Response

Response: Thank you for the positive comments. The time and the valuable comments of the reviewers much appreciated.

Point 1. The quality of figure 3 is too low: it is impossible to read its content! The authors are invited to fix that issue.

Response 1: Figure 3 has been replaced with a better version.

Point 2. Paragraphs 3.4 and 4.1 should be combined together in a single paragraph.

Response 2. While this is a very valid point, section 3 discusses the types of polymers used in mineral and metallurgical processing. Section 4 discusses the actual applications. The super absorbent polymers discussed in 3.4 are used in coal-dewatering applications in 4.1 hence closely related. However, merging these two sections will alter the entire structure of the paper.

Point 3. Authors are invited to check and standardize text styles throughout the paper.

Response 3. We have checked and made some corrections